# Brain-derived estrogens facilitate male-typical behaviors by potentiating androgen receptor signaling in medaka

Yuji Nishiike[1], Shizuku Maki[1], Daichi Miyazoe[1], Kiyoshi Nakasone[1], Yasuhiro Kamei[2], Takeshi Todo[3], Tomoko Ishikawa-Fujiwara[3], Kaoru Ohno[4], Takeshi Usami[4], Yoshitaka Nagahama[4], Kataaki Okubo[1]*

[1]Department of Aquatic Bioscience, Graduate School of Agricultural and Life Sciences, The University of Tokyo, Bunkyo, Japan; [2]Optics and Bioimaging Facility, Trans-Scale Biology Center, National Institute for Basic Biology, Okazaki, Japan; [3]Department of Genome Biology, Graduate School of Medicine, Osaka University, Suita, Japan; [4]Division of Reproductive Biology, National Institute for Basic Biology, Okazaki, Japan

*For correspondence:
a-okubo@g.ecc.u-tokyo.ac.jp

## eLife Assessment

This is an overall **compelling** set of findings on the role of centrally produced estrogens in the control of behaviors in male medaka. The significance of the findings rests on the revealed potential mechanism between brain derived estrogens modulating social behaviors in males, supported by the analysis of multiple transgenic lines. The evidence for the broader claim is incomplete since it has not been extended to female medaka, and further experimentation would be necessary to fully validate the conclusions on the role of brain-derived estrogens. Nonetheless, the findings have led to **important** hypotheses on the hormonal control of behaviors in teleosts that can be tested further.

**Abstract** In rodents, estrogens aromatized from androgens in the brain are essential for the development of male-typical behaviors. In many other vertebrates, including humans and teleost fish, however, androgens facilitate these behaviors directly via the androgen receptor without aromatization into estrogens. Here, we report that mutagenesis-derived male medaka fish lacking Cyp19a1b (a subtype of aromatase predominantly expressed in the brain) exhibit severely impaired male-typical mating and aggression, despite elevated brain androgen levels. These phenotypes can be rescued by estrogen administration, indicating that brain-derived estrogens are pivotal for male-typical behaviors even in teleosts. Our results further suggest that these estrogens facilitate male-typical behaviors by potentiating androgen action in the brain via the direct stimulation of androgen receptor transcription. Taken together, these findings reveal a previously unappreciated mode of action of brain-derived estrogens in facilitating male-typical behaviors.

## Introduction

Male and female animals exhibit differences in many innate behaviors, such as mating and aggression (*Zilkha et al., 2021*). In vertebrates, these differences are driven primarily by the influence of sex steroid hormones, including estrogens and androgens. Extensive research in rodents has established that estrogens, traditionally considered 'female' hormones, are critical for the development of male-typical behaviors (*Ogawa et al., 2020*; *Tsukahara and Morishita, 2020*; *McCarthy, 2023*). Specifically, androgens secreted by the testis, both perinatally and in adulthood, are converted to

estrogens in the brain by the enzyme aromatase. These estrogens then act through ESR1, a subtype of the estrogen receptor (ESR), to elicit male-typical behaviors (*Ogawa et al., 2020*; *Tsukahara and Morishita, 2020*; *McCarthy, 2023*). This process, originally referred to as the 'aromatization hypothesis', is now widely acknowledged; however, the mechanisms through which brain-derived estrogens affect male-typical behavior, including the identity of their downstream targets, remain largely elusive (*McCarthy et al., 2017*; *McCarthy, 2023*).

More importantly, the aromatization hypothesis seems to apply to only a limited number of species besides rodents (e.g. some birds such as zebra finches) (*Balthazart, 2019*). In other species such as humans, other primates, and teleost fish, testicular androgens facilitate male-typical behaviors directly through the androgen receptor (AR) without aromatization (*Thornton et al., 2009*; *Bakker, 2022*; *Okubo et al., 2022*). In primates, the hypothalamic aromatization of androgens to estrogens plays a central role in female gametogenesis (*Terasawa, 2018*) but is not essential for male behaviors (*Thornton et al., 2009*; *Bakker, 2022*). Notably, in teleosts, 11-ketotestosterone (11KT), which cannot be aromatized to estrogens, is the primary testicular androgen, and exogenous 11KT effectively induces male-typical courtship and aggression even in females (*Nishiike et al., 2021*; *Okubo et al., 2022*; *Kawabata-Sakata et al., 2024*). Therefore, it is generally assumed that, in teleosts, androgens are largely responsible for male-typical behaviors, while estrogens are dispensable for these behaviors. This is consistent with recent observations in a few teleost species that genetic ablation of AR severely impairs male-typical behaviors (*Yong et al., 2017*; *Alward et al., 2020*; *Ogino et al., 2023*; *Nishiike and Okubo, 2024*) and with findings in medaka fish (*Oryzias latipes*) that estrogens act through the ESR subtype Esr2b to prevent females from engaging in male-typical courtship (*Nishiike et al., 2021*).

Interestingly, despite these findings, adult teleost brains have extremely high levels of aromatase activity (100–1000 times higher than in rodent brains), resulting in large amounts of brain estrogens even in males (*Diotel et al., 2018*). In teleost brains, including those of medaka, aromatase is exclusively localized in radial glial cells, in contrast to its neuronal localization in rodent brains (*Forlano et al., 2001*; *Diotel et al., 2010*; *Takeuchi and Okubo, 2013*). These observations suggest that brain-derived estrogens have a vital, but as yet undetermined, role in male teleosts. It is worth mentioning that systemic administration of estrogens and an aromatase inhibitor increased and decreased male aggression, respectively, in several teleost species, potentially reflecting the behavioral effects of brain-derived estrogens (*Hallgren et al., 2006*; *O'Connell and Hofmann, 2012*; *Huffman et al., 2013*; *Jalabert et al., 2015*). Most teleost species have two distinct genes encoding aromatase, *cyp19a1a* and *cyp19a1b*, due to a whole-genome duplication that occurred early in teleost evolution (*Diotel et al., 2018*; *Nagahama et al., 2021*). These genes have undergone subfunctionalization through the partitioning of tissue-specific expression patterns: *cyp19a1a* is expressed predominantly in the gonad, whereas *cyp19a1b* is expressed in the brain (*Diotel et al., 2018*; *Nagahama et al., 2021*). In medaka, *cyp19a1b* is also expressed in the gonads, but only at a level tens to hundreds of times lower than in the brain and substantially lower than that of *cyp19a1a* (*Okubo et al., 2011*; *Nakamoto et al., 2018*).

In the present study, we generated *cyp19a1b*-deficient medaka, in which estrogen synthesis in the brain is selectively impaired while that in the gonads remains intact, in order to investigate the impact of brain-derived estrogens on male behaviors. Remarkably, the fish showed severely impaired male-typical behaviors, thus revealing the marked behavioral effects of these estrogens. Our results further suggest that these effects are mediated by potentiating androgen signaling in behaviorally relevant brain regions.

## Results

### *cyp19a1b*-deficient males exhibit severely impaired male-typical mating and aggressive behaviors

We generated a *cyp19a1b*-deficient medaka line from a mutant founder carrying a nonsense mutation in exon 4 of *cyp19a1b* that was identified through screening of the medaka TILLING (targeting-induced local lesions in genomes) library (*Taniguchi et al., 2006*; *Figure 1—figure supplement 1A and B*). Loss of *cyp19a1b* function in these fish was verified by measuring brain and peripheral levels of sex steroids in males. As expected, brain estradiol-17β (E2) in homozygous mutants (*cyp19a1b$^{-/-}$*) was significantly reduced to 16% of the levels in wild-type (*cyp19a1b$^{+/+}$*) siblings (p=0.0037) (*Figure 1A*). Brain E2 in

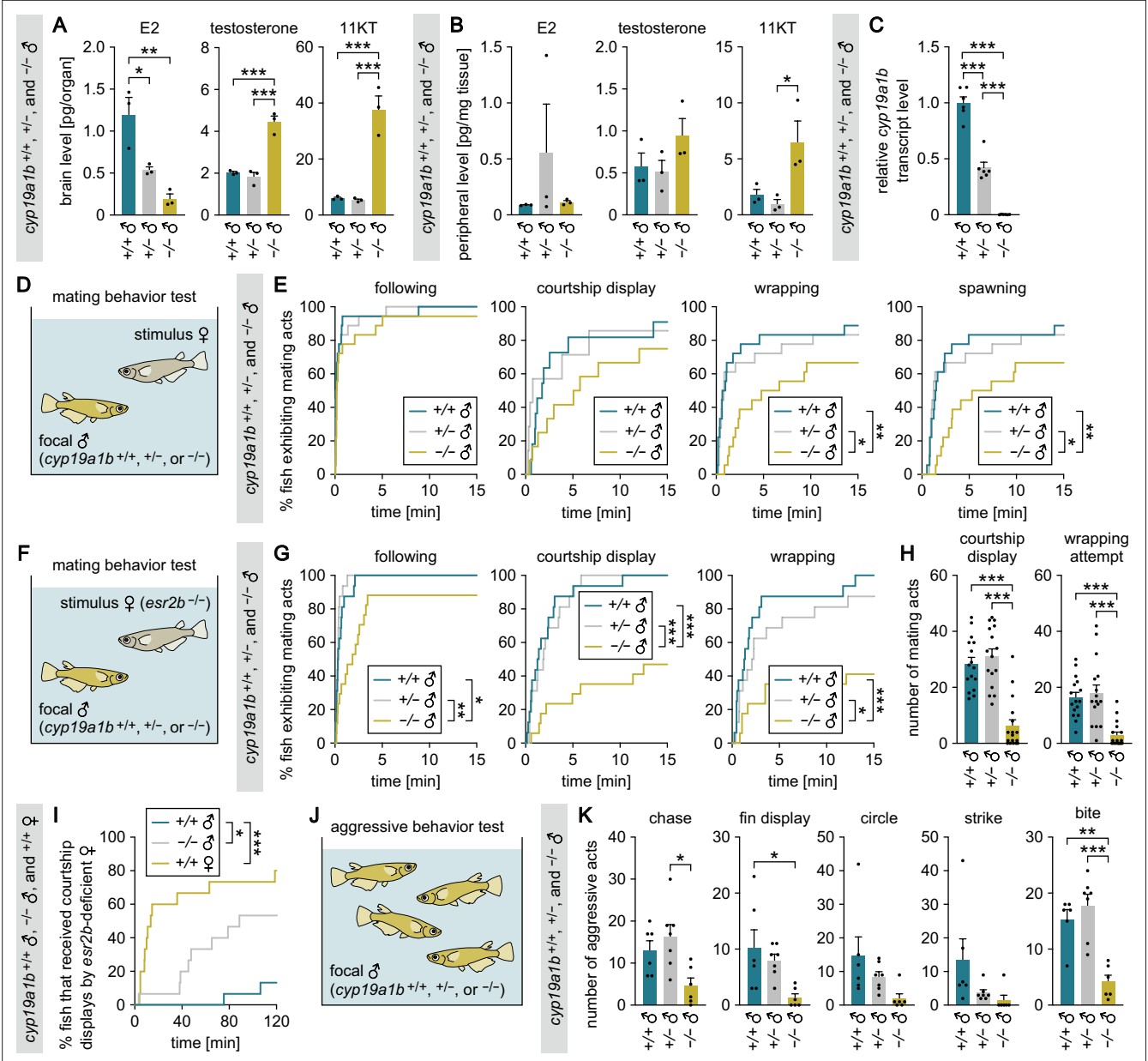

Figure 1. *cyp19a1b*-deficient males exhibit severely impaired male-typical mating and aggressive behaviors. (**A, B**) Levels of E2, testosterone, and 11-ketotestosterone (11KT) in the brain (**A**) and periphery (**B**) of adult *cyp19a1b*[+/+], *cyp19a1b*[+/−], and *cyp19a1b*[−/−] males (n=3 per genotype). (**C**) Brain *cyp19a1b* transcript levels in *cyp19a1b*[+/+], *cyp19a1b*[+/−], and *cyp19a1b*[−/−] males (n=6 per genotype). Mean value for *cyp19a1b*[+/+] males was arbitrarily set to 1. (**D**) Setup for testing the mating behavior of *cyp19a1b*[+/+], *cyp19a1b*[+/−], and *cyp19a1b*[−/−] males. (**E**) Latency of *cyp19a1b*[+/+], *cyp19a1b*[+/−], and *cyp19a1b*[−/−] males (n=18 per genotype) to initiate each mating act toward the stimulus female. (**F**) Setup for testing mating behavior using an *esr2b*-deficient female as the stimulus. (**G**) Latency of *cyp19a1b*[+/+], *cyp19a1b*[+/−], and *cyp19a1b*[−/−] males (n=16, 16, and 17, respectively) to initiate each mating act toward the *esr2b*-deficient female. (**H**) Number of each mating act performed. (**I**) Latency of *cyp19a1b*[+/+] and *cyp19a1b*[−/−] males and *cyp19a1b*[+/+] females (n=15 each) to receive courtship displays from the *esr2b*-deficient female. (**J**) Setup for testing aggressive behavior among grouped males. (**K**) Total number of each aggressive act performed by *cyp19a1b*[+/+], *cyp19a1b*[+/−], and *cyp19a1b*[−/−] males. Each data point represents the sum of acts recorded for the 4 males of the same genotype in a single tank (n=6, 7, and 6 tanks, respectively). Statistical differences were assessed by Bonferroni's or Dunn's post hoc test (**A, B, C, H, K**) and Gehan-Breslow-Wilcoxon test with Bonferroni's correction (**E, G, I**). Error bars represent SEM. *$p < 0.05$, **$p < 0.01$, ***$p < 0.001$.

The online version of this article includes the following source data and figure supplement(s) for figure 1:

**Source data 1.** Source data for *Figure 1*.

**Figure supplement 1.** Generation of *cyp19a1b*-deficient medaka.

heterozygotes (*cyp19a1b*$^{+/-}$) was also reduced to 45% of wild-type levels (p=0.0284) (*Figure 1A*), indicating a dosage effect of the *cyp19a1b* mutation. In contrast, peripheral E2 levels were unaltered in *cyp19a1b*$^{-/-}$ males (*Figure 1B*), consistent with the expected functioning of Cyp19a1b primarily in the brain. Strikingly, brain testosterone levels, as opposed to E2, increased 2.2-fold in *cyp19a1b*$^{-/-}$ males relative to wild-type siblings (p=0.0006) (*Figure 1A*). Similarly, brain 11KT levels increased 6.2-fold (p=0.0007) (*Figure 1A*). These results indicate that *cyp19a1b*-deficient males have reduced estrogen coupled with elevated androgen levels in the brain, confirming the loss of *cyp19a1b* function. They also suggest that the majority of estrogens in the male brain are synthesized locally in the brain. Peripheral 11KT levels also increased 3.7-fold in *cyp19a1b*$^{-/-}$ males (p=0.0789) (*Figure 1B*), indicating peripheral influence in addition to central effects.

Loss of *cyp19a1b* function was further confirmed by measuring *cyp19a1b* transcript levels in the brain and by predicting the three-dimensional structure of the mutant protein. Real-time PCR revealed that transcript levels were reduced by half in *cyp19a1b*$^{+/-}$ males and were nearly undetectable in *cyp19a1b*$^{-/-}$ males, presumably as a result of nonsense-mediated mRNA decay (*Lindeboom et al., 2019*; *Figure 1C*). The wild-type protein, modeled by AlphaFold 3, exhibited a typical cytochrome P450 fold, including the membrane helix, aromatic region, and heme-binding loop, all arranged in the expected configuration (*Figure 1—figure supplement 1C*). The mutant protein, in contrast, was severely truncated, retaining only the membrane helix (*Figure 1—figure supplement 1C*). The absence of essential domains strongly indicates that the allele encodes a nonfunctional Cyp19a1b protein. Together, transcript and structural analyses consistently demonstrate that the mutation generated in this study causes a complete loss of *cyp19a1b* function.

Next, we investigated the mating behavior of *cyp19a1b*-deficient males (*Figure 1D*). The mating behavior of medaka follows a stereotypical sequence. It begins with the male approaching and closely following the female (following). The male then performs a courtship display, rapidly swimming in a circular pattern in front of the female. If the female is receptive, the male grasps her with his fins (wrapping), culminating in the simultaneous release of eggs and sperm (spawning) (*Nishiike and Okubo, 2024*). Because 11KT, the primary driver of male-typical behaviors in teleosts, was increased in *cyp19a1b*-deficient males, we predicted that these fish would engage more actively in mating. Nevertheless, *cyp19a1b*$^{-/-}$ males showed significantly longer latencies to initiate wrappings (p=0.0033 versus *cyp19a1b*$^{+/+}$, p=0.0195 versus *cyp19a1b*$^{+/-}$) and to spawn (p=0.0051 versus *cyp19a1b*$^{+/+}$, p=0.0195 versus *cyp19a1b*$^{+/-}$) (*Figure 1E*). These results suggest that they are less motivated to mate, though an alternative interpretation that their cognition or sexual preference may be altered cannot be dismissed.

However, no significant differences were evident in latencies to initiate followings and courtship displays (*Figure 1E*); therefore, the possibility that *cyp19a1b*-deficient males are less sexually attractive and less preferred by females could not be ruled out. To ascertain whether *cyp19a1b*-deficient males are indeed less motivated to mate, we further tested their mating behavior using *esr2b*-deficient females, which are unreceptive to male courtship (*Nishiike et al., 2021*), as stimulus females (*Figure 1F*); this test eliminates the influence of female receptivity, facilitating an exclusive evaluation of male motivation to mate with females (*Nishiike and Okubo, 2024*). We found that *cyp19a1b*$^{-/-}$ males showed significantly longer latencies to initiate followings (p=0.0129 versus *cyp19a1b*$^{+/+}$, p=0.0060 versus *cyp19a1b*$^{+/-}$), courtship displays (p=0.0003 versus *cyp19a1b*$^{+/+}$, p=0.0006 versus *cyp19a1b*$^{+/-}$), and wrapping attempts (p=0.0009 versus *cyp19a1b*$^{+/+}$, p=0.0282 versus *cyp19a1b*$^{+/-}$) (*Figure 1G*). In addition, they exhibited significantly fewer courtship displays (p<0.0001 versus both *cyp19a1b*$^{+/+}$ and *cyp19a1b*$^{+/-}$) and wrapping attempts (p<0.0001 versus both *cyp19a1b*$^{+/+}$ and *cyp19a1b*$^{+/-}$) (*Figure 1H*). These results thus confirmed that *cyp19a1b*-deficient males are less motivated to mate. In our previous study, we found that *esr2b*-deficient females court females preferentially over males (*Nishiike et al., 2021*). Consistent with that finding, here, we observed that 12 of 15 *cyp19a1b*$^{+/+}$ females received courtship displays from the *esr2b*-deficient female, as compared with only 2 of 15 *cyp19a1b*$^{+/+}$ males (p=0.0003) (*Figure 1I*). Curiously, we further observed that 8 of 15 *cyp19a1b*$^{-/-}$ males received courtship displays from the *esr2b*-deficient female (p=0.0321 versus *cyp19a1b*$^{+/+}$ males) (*Figure 1I*). Perhaps *cyp19a1b*$^{-/-}$ males are misidentified as females by *esr2b*-deficient females because they are reluctant to court or they exhibit some female-like behavior.

Next, we examined the intrasexual aggressive behavior of *cyp19a1b*-deficient males (*Figure 1J*). The aggressive behavior of teleosts including medaka consists of five types of behavioral acts: chases,

fin displays, circles, strikes, and bites (*Yamashita et al., 2020*). *cyp19a1b*⁻/⁻ males exhibited all of these aggressive acts less frequently than *cyp19a1b*⁺/⁺ and/or *cyp19a1b*⁺/⁻ males, with significant differences observed for chases (p=0.0123 versus *cyp19a1b*⁺/⁻), fin displays (p=0.0214 versus *cyp19a1b*⁺/⁺), and bites (p=0.0015 versus *cyp19a1b*⁺/⁺, p=0.0002 versus *cyp19a1b*⁺/⁻) (*Figure 1K*). These observations demonstrate that *cyp19a1b*-deficient males are less aggressive toward other males.

In tilapia (*Oreochromis niloticus*), depletion of *cyp19a1b* has been reported to cause male infertility due to efferent duct obstruction (*Zhang et al., 2019*). If this is also the case in medaka, the observed behavioral defects might be secondary to infertility, possibly due to the perception of impaired sperm release. We therefore investigated the fertilization and hatching success of embryos derived from mating between *cyp19a1b*⁻/⁻ males and wild-type females. Their fertilization and hatching rates were 88.5% (n=183) and 93.2% (n=162), respectively, indicating that *cyp19a1b*-deficient male medaka have normal fertility.

## Brain-derived estrogens facilitate male-typical behaviors probably by stimulating brain AR expression

We considered that the impaired male-typical behaviors of *cyp19a1b*-deficient males might be reasonably attributed to either reduced E2 or increased 11KT in the brain. The latter possibility seemed unlikely because 11KT/AR signaling strongly promotes male-typical behaviors in teleosts (*Yong et al., 2017*; *Alward et al., 2020*; *Okubo et al., 2022*; *Ogino et al., 2023*; *Nishiike and Okubo, 2024*); therefore, we tested the former possibility by examining whether E2 treatment would rescue the behavioral phenotypes of *cyp19a1b*-deficient males (*Figure 2A*). E2 treatment, while having no effect in *cyp19a1b*⁺/⁺ males (*Figure 2B and C*), significantly shortened the latency to (p=0.0005) and increased the number of (p=0.0006) courtship displays in *cyp19a1b*⁻/⁻ males (*Figure 2D and E*). These results suggest that reduced E2 in the brain is the primary cause of the mating defects, highlighting a pivotal role of brain-derived estrogens in male mating behavior. However, caution is warranted, as an indirect peripheral effect of bath-immersed E2 on behavior cannot be ruled out, although this is unlikely given the comparable peripheral E2 levels in *cyp19a1b*-deficient and wild-type males. In contrast to mating, E2 treatment was not effective in restoring aggression in *cyp19a1b*⁻/⁻ males (*Figure 2—figure supplement 1A*). It is possible that the treatment protocol used may have failed to replicate the estrogenic milieu necessary to induce aggression in males.

We then considered the mechanism by which brain-derived estrogens facilitate male-typical behaviors. Our previous study showed that exogenous E2 upregulates the expression of a subtype of AR, Ara, in the medaka brain (*Hiraki et al., 2012*; note that Ara was termed Arb in this reference). We therefore hypothesized that brain-derived estrogens may facilitate male-typical behaviors by stimulating Ara expression and thereby potentiating 11KT/Ara signaling in the brain. To test this hypothesis, we first examined *ara* expression in the brain of *cyp19a1b*-deficient males by in situ hybridization analysis. Expression of *ara* was significantly lower in *cyp19a1b*⁻/⁻ males than in *cyp19a1b*⁺/⁺ males in several preoptic and hypothalamic nuclei that are activated upon mating and/or attack in males (*Nishiike and Okubo, 2024*), including the PPa, pPPp, and NVT (p=0.0134, 0.0372, and 0.0008, respectively) (*Figure 2F and G*, *Supplementary file 1* for abbreviations of brain nuclei). We then performed a similar analysis for the other AR subtype, Arb. Expression of *arb* was significantly lower in *cyp19a1b*⁻/⁻ than in *cyp19a1b*⁺/⁺ males in other preoptic and hypothalamic nuclei activated upon mating and/or attack (*Nishiike and Okubo, 2024*), including the PMp, aPPp, and NPT (p=0.0009, 0.0413, and 0.0021, respectively) (*Figure 2H and I*).

Next, to determine whether these reductions in *ara* and *arb* expression in *cyp19a1b*⁻/⁻ males were the result of reduced brain E2, we tested whether E2 treatment could restore the expression of the two genes. In situ hybridization revealed significantly increased *ara* and *arb* expression in most brain nuclei of E2-treated *cyp19a1b*⁻/⁻ males as compared with vehicle-treated *cyp19a1b*⁻/⁻ males (similar results were obtained in *cyp19a1b*⁺/⁺ males) (*Figure 2J and K*, *Figure 2—figure supplement 1B and C*, *Figure 2—figure supplement 2*), indicating that the decreased *ara* and *arb* expression in *cyp19a1b*⁻/⁻ males is attributable to reduced E2 levels. Taken together, these results suggest that brain-derived estrogens elicit male-typical behaviors by stimulating *ara* and *arb* expression in behaviorally relevant brain regions.

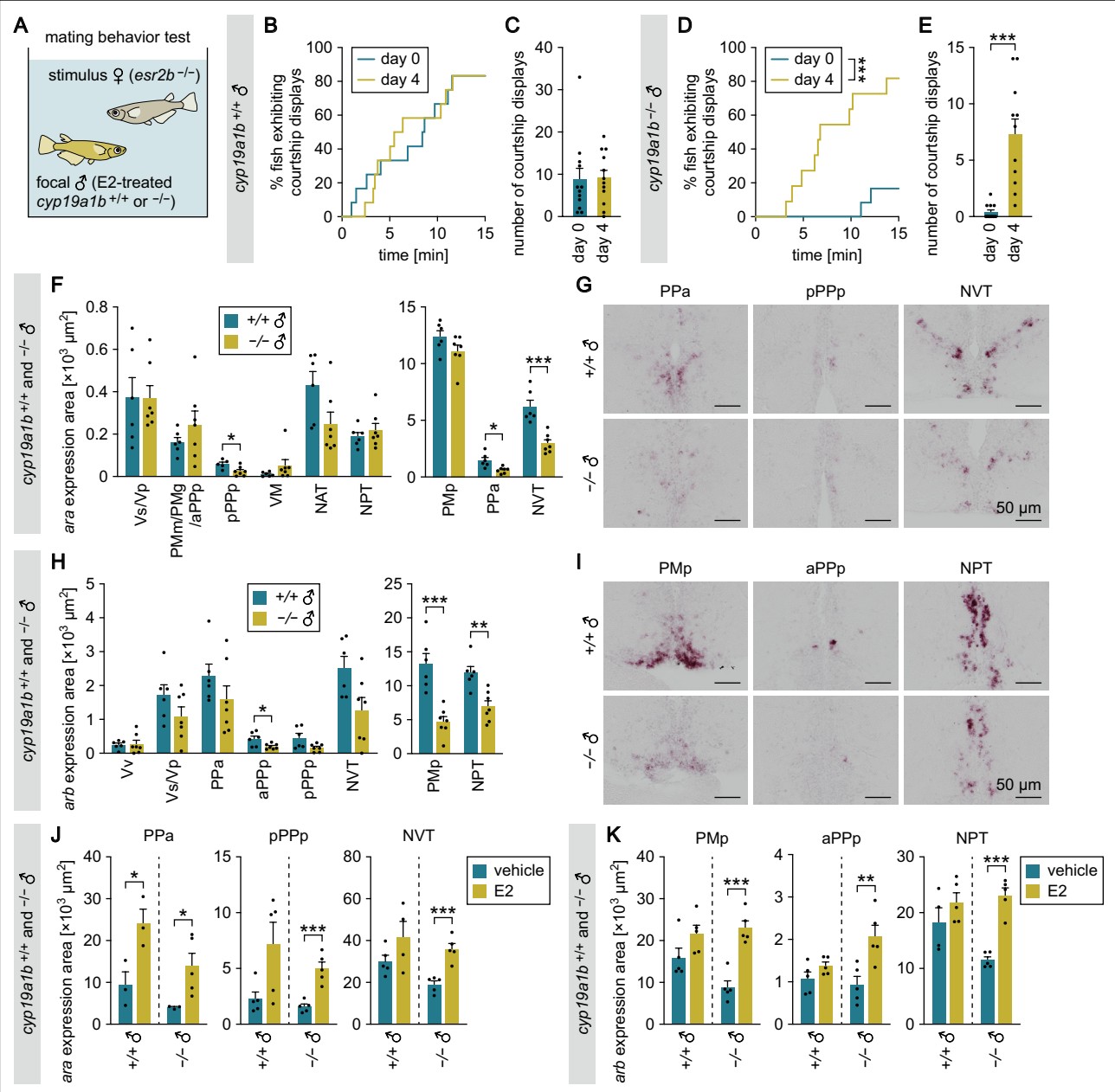

**Figure 2.** Brain-derived estrogens facilitate male-typical behaviors probably by stimulating brain AR expression. (**A**) Setup for testing the mating behavior of E2-treated *cyp19a1b*<sup>+/+</sup> and *cyp19a1b*<sup>−/−</sup> males. (**B**) Latency of *cyp19a1b*<sup>+/+</sup> males (n=12) to initiate courtship displays toward the stimulus female before (day 0) and after (day 4) E2 treatment. (**C**) Number of courtship displays performed by *cyp19a1b*<sup>+/+</sup> males. (**D**) Latency of *cyp19a1b*<sup>−/−</sup> males to initiate courtship displays before (day 0; n=12) and after (day 4; n=11) E2 treatment. (**E**) Number of courtship displays performed by *cyp19a1b*<sup>−/−</sup> males. (**F**) Total area of *ara* expression signal in each brain nucleus of *cyp19a1b*<sup>+/+</sup> (n=6 except for pPPp, where n=5) and *cyp19a1b*<sup>−/−</sup> (n=7) males. The data are displayed in two graphs for visual clarity. (**G**) Representative images of *ara* expression in the PPa, pPPp, and NVT. (**H**) Total area of *arb* expression signal in each brain nucleus of *cyp19a1b*<sup>+/+</sup> (n=6) and *cyp19a1b*<sup>−/−</sup> (n=7) males. The data are displayed in two graphs for visual clarity. (**I**) Representative images of *arb* expression in the PMp, aPPp, and NPT. (**J**) Total area of *ara* expression signal in the PPa, pPPp, and NVT of *cyp19a1b*<sup>+/+</sup> and *cyp19a1b*<sup>−/−</sup> males treated with vehicle alone or E2 (n=5 per group except for NVT of E2-treated *cyp19a1b*<sup>+/+</sup> males, where n=4; and PPa of vehicle-treated *cyp19a1b*<sup>+/+</sup>, E2-treated *cyp19a1b*<sup>+/+</sup>, and vehicle-treated *cyp19a1b*<sup>−/−</sup> males, where n=3). (**K**) Total area of *arb* expression signal in the PMp, aPPp, and NPT of *cyp19a1b*<sup>+/+</sup> and *cyp19a1b*<sup>−/−</sup> males treated with vehicle alone or E2 (n=5 per group except for NPT of vehicle-treated *cyp19a1b*<sup>+/+</sup> males, where n=4). Scale bars represent 50 µm. For abbreviations of brain nuclei, see **Supplementary file 1**. Statistical differences were assessed by Gehan-Breslow-Wilcoxon test (**B, D**) and unpaired *t* test, with Welch's correction where appropriate (**C, E, F, H, J, K**). Error bars represent SEM. *p<0.05, **p<0.01, ***p<0.001.

The online version of this article includes the following source data and figure supplement(s) for figure 2:

*Figure 2 continued on next page*

*Figure 2 continued*

**Source data 1.** Source data for *Figure 2*.

**Figure supplement 1.** Effect of estrogen replacement on aggression and brain *ara* and *arb* expression in *cyp19a1b*-deficient males.

**Figure supplement 1—source data 1.** Source data for *Figure 2—figure supplement 1*.

**Figure supplement 2.** Effect of estrogen replacement on brain *ara* and *arb* expression in *cyp19a1b*-deficient males.

### *cyp19a1b* deficiency impairs behaviorally relevant signaling pathways downstream of ARs

We considered that, if our hypothesis that brain-derived estrogens facilitate male-typical behaviors by potentiating androgen/AR signaling is correct, then *cyp19a1b*-deficient males should have impaired activation of behaviorally relevant genes that act downstream of ARs. Two neuropeptide genes, *vt* (encoding vasotocin) and *gal* (encoding galanin), have been implicated in male-typical mating and aggressive behaviors in various vertebrates, including medaka and other teleosts (*Yokoi et al., 2015*; *Tripp et al., 2020*; *Yamashita et al., 2020*; *Kawabata-Sakata et al., 2024*; *Nishiike and Okubo, 2024*). In medaka, subsets of neurons in the pNVT and pPMp express *vt* and *gal*, respectively, in an androgen-dependent and hence male-biased manner, and these neurons have been implicated in male-typical behaviors (*Kawabata et al., 2012*; *Yamashita et al., 2020*; *Kawabata-Sakata et al., 2024*). We therefore studied the expression of *vt* and *gal* in the brains of *cyp19a1b*-deficient males.

In situ hybridization revealed that, as expected, expression of *vt* in the pNVT of *cyp19a1b⁻/⁻* males was significantly reduced to 18% as compared with *cyp19a1b⁺/⁺* males (p=0.0040) (*Figure 3A and B*). In contrast, there were no significant differences between genotypes in other brain nuclei (*Figure 3A*). Similarly, expression of *gal* in the pPMp of *cyp19a1b⁻/⁻* males was reduced to 43% as compared with *cyp19a1b⁺/⁺* males (p=0.0019), while no significant differences were observed in other brain nuclei (*Figure 3C and D*). These results demonstrate that *cyp19a1b* deficiency severely impairs the AR-dependent activation of behaviorally relevant *vt* and *gal* expression, suggesting that brain-derived estrogens play a substantial role in activating AR signaling.

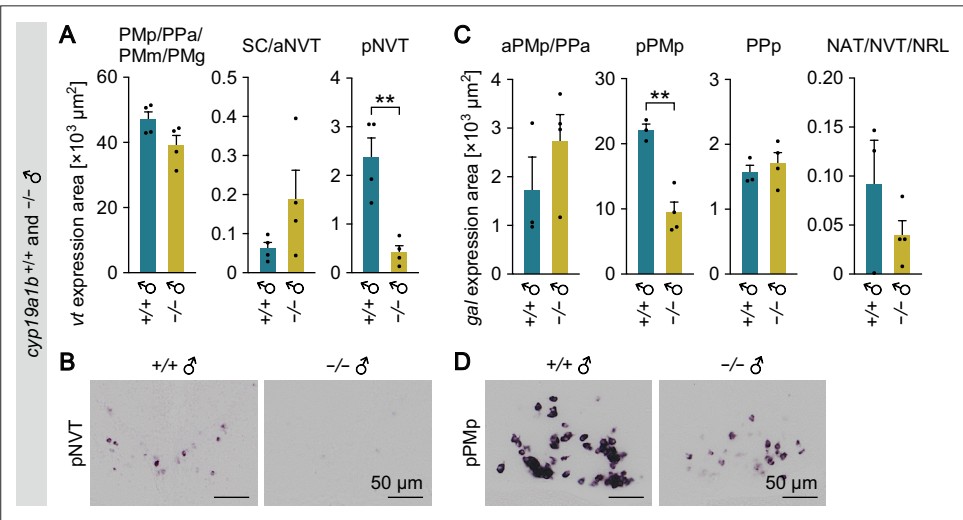

**Figure 3.** *cyp19a1b* deficiency impairs behaviorally relevant signaling pathways downstream of ARs. (**A**) Total area of *vt* expression signal in the PMp/PPa/PMm/PMg, SC/aNVT, and pNVT of *cyp19a1b⁺/⁺* and *cyp19a1b⁻/⁻* males (n=4 per genotype). (**B**) Representative images of *vt* expression in the pNVT. (**C**) Total area of *gal* expression signal in the aPMp/PPa, pPMp, PPp, and NAT/NVT/NRL of *cyp19a1b⁺/⁺* and *cyp19a1b⁻/⁻* males (n=3 and 4, respectively). (**D**) Representative images of *gal* expression in the pPMp. Scale bars represent 50 μm. For abbreviations of brain nuclei, see *Supplementary file 1*. Statistical differences were assessed by unpaired *t* test, with Welch's correction where appropriate (**A, C**). Error bars represent SEM. **p<0.01.

The online version of this article includes the following source data for figure 3:

**Source data 1.** Source data for *Figure 3*.

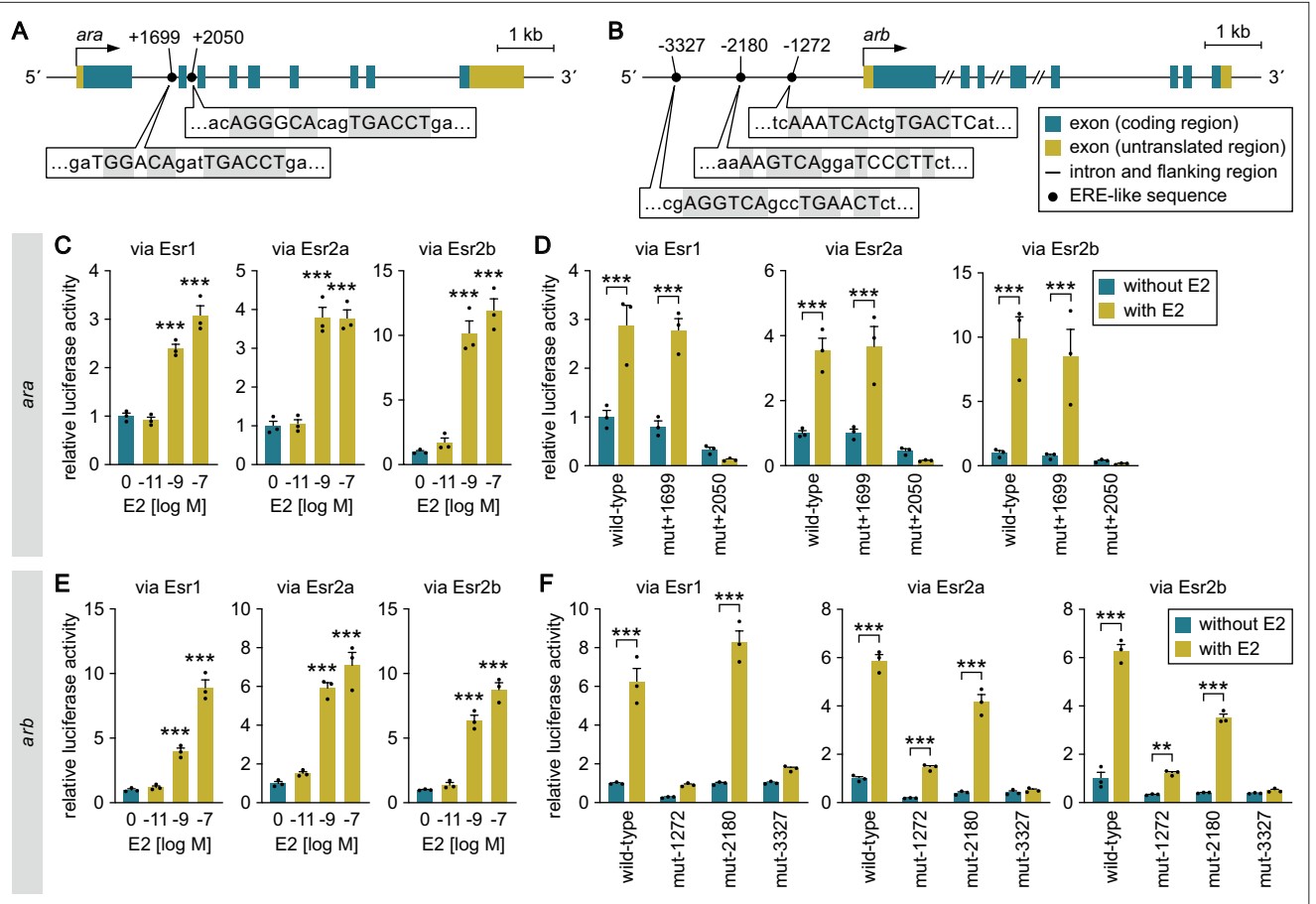

**Figure 4.** Estrogens directly stimulate the transcription of ARs through ESRs. (**A, B**) Schematic of *ara* (**A**) and *arb* (**B**) loci showing the location of the canonical bipartite ERE-like sequences. Bent arrows mark the transcription initiation sites. Nucleotides of the ERE-like sequences are denoted by capital letters, and those identical to the consensus ERE (AGGTCAnnnTGACCT) are gray-shaded. (**C**) Ability of E2 to directly activate *ara* transcription. Cultured cells were transfected with a luciferase reporter construct containing a genomic fragment upstream of exon 3 of *ara*, together with an Esr1, Esr2a, or Esr2b expression construct. The cells were stimulated with different concentrations of E2, and luciferase activity was measured. (**D**) Effect of mutations in the ERE-like sequences on the E2-induced activation of *ara* transcription. Cultured cells were transfected with a wild-type luciferase construct or a construct carrying a mutation in the ERE-like sequence at position +1699 (mut +1699) or +2050 (mut+2050), together with an Esr1, Esr2a, or Esr2b expression construct. The cells were stimulated with or without E2, and luciferase activity was measured. (**E**) Ability of E2 to directly activate *arb* transcription. The assay described in C was performed with a luciferase construct containing a genomic fragment upstream of the first methionine codon of *arb*. (**F**) Effect of mutations in the ERE-like sequences on the E2-induced activation of *arb* transcription. The assay described in D was performed with luciferase constructs, each carrying a mutation in the ERE-like sequence at position –1272 (mut–1272), –2180 (mut–2180), or –3327 (mut–3327). Values are expressed as a fold change relative to a control without E2 stimulation (**C, E**) or a control using the wild-type construct without E2 stimulation (**D, F**). Statistical differences were assessed by Dunnett's post hoc test (**C, E**) and unpaired *t* test with Bonferroni-Dunn correction (**D, F**). Error bars represent SEM. **$p<0.01$; ***$p<0.001$.

The online version of this article includes the following source data and figure supplement(s) for figure 4:

**Source data 1.** Source data for *Figure 4*.

**Figure supplement 1.** Effect of mutations in each half-site of the identified estrogen-responsive element (ERE) on the E2-induced activation of *ara* and *arb* transcription.

**Figure supplement 1—source data 1.** Source data for *Figure 4—figure supplement 1*.

## Estrogens directly stimulate the transcription of ARs through ESRs

The above results led us to further explore how brain-derived estrogens stimulate the expression of *ara* and *arb*. In silico analysis of the medaka *ara* and *arb* loci identified two canonical bipartite estrogen-responsive element (ERE)-like sequences in introns 1 and 2 of *ara* (located at positions +1699 and +2050, respectively, relative to the transcription initiation site) and three in the 5′-flanking region of *arb* (at positions –1272, –2180, and –3327) (*Figure 4A and B*). Thus, brain estrogens might

be able to directly activate the transcription of *ara* and *arb*. To evaluate this likelihood, we conducted in vitro transcriptional activity assays in which luciferase expression was driven by genomic fragments from the *ara* and *arb* loci containing the identified ERE-like sequences.

An assay using the *ara* genomic fragment revealed that E2 dose-dependently increased luciferase activity in the presence of any ESR subtype (Esr1, Esr2a, or Esr2b) (*Figure 4C*), suggesting that E2 has the ability to directly activate *ara* transcription through ESRs. The introduction of a point mutation into the ERE-like sequence at position +2050 abolished the E2-induced luciferase activity in the presence of any ESR subtype, while mutation at +1699 had no such effect (*Figure 4D*). These results suggest that the ERE at position +2050 is responsible for E2-induced activation of *ara* transcription. Because there is evidence that a single ERE half-site is sufficient to confer E2 responsiveness on several genes (*Klinge, 2001*), we performed the assay using *ara* genomic fragments in which a mutation was introduced in either half-site of the ERE at +2050. E2 induction of luciferase activity was abrogated in both cases (*Figure 4—figure supplement 1A*), suggesting that both ERE half-sites are required to confer estrogen responsiveness on *ara*.

In the assay using the *arb* genomic fragment, E2 also dose-dependently increased luciferase activity in the presence of any ESR subtype (*Figure 4E*). The E2-induced increase was completely abolished by point mutation of the ERE-like sequence at position –3327 in the presence of any ESR subtype (*Figure 4F*). A similar effect was observed for mutation of the ERE-like sequence at position –1272, but only in the presence of Esr1 and not in the presence of Esr2a or Esr2b (*Figure 4F*). These results indicate that E2 can directly stimulate *arb* transcription, primarily through the ERE at –3327. Mutations in either half-site of this ERE eliminated induction by E2 in the presence of Esr2a or Esr2b, but not Esr1 (*Figure 4—figure supplement 1B*), suggesting that both ERE half-sites are required to confer estrogen responsiveness on *arb*. Collectively, these experiments suggest that the transcription of *ara* and *arb* can be directly stimulated by estrogens (including brain-derived estrogens) via the binding of ESRs to canonical bipartite EREs.

## Brain-derived estrogens stimulate *ara* and *arb* expression in behaviorally relevant brain regions primarily through Esr2a and Esr1, respectively

Next, we investigated whether brain-derived estrogens can induce *ara* and *arb* expression through ESRs in vivo and, if so, which ESR subtype (Esr1, Esr2a, or Esr2b) mediates this induction. To this end, we examined *ara* and *arb* expression in the brains of males deficient for each ESR subtype by in situ hybridization. We used previously described *esr1*- and *esr2b*-deficient medaka (*Nishiike et al., 2021*; *Fleming et al., 2023*) and generated *esr2a*-deficient medaka using the CRISPR (clustered regularly interspaced short palindromic repeats)/Cas9 (CRISPR-associated protein 9) system. Two independent *esr2a*-deficient lines (Δ8 and Δ4) were established (*Figure 5—figure supplement 1*) and used for subsequent behavioral experiments to ensure the reproducibility of the observed phenotypes.

In $esr1^{+/+}$ and $esr1^{-/-}$ males, *ara* expression was not significantly different between genotypes in any brain nucleus (*Figure 5—figure supplement 2A*), but *arb* expression was significantly lower in $esr1^{-/-}$ than in $esr1^{+/+}$ males in the PMp and aPPp (p=0.0111 and 0.0376, respectively) (*Figure 5A and B*). This, together with our observation that *arb* expression in these brain nuclei was also significantly lower in $cyp19a1b^{-/-}$ males (*Figure 2H*), suggests that brain-derived estrogens stimulate *arb* expression in these nuclei primarily through Esr1. In support of this notion, double-label in situ hybridization in the wild-type male brain detected neurons coexpressing *arb* and *esr1* in both the PMp and aPPp (*Figure 5—figure supplement 2B*). In $esr2a^{+/+}$ and $esr2a^{-/-}$ male brains (from the Δ8 line), *ara* expression was significantly lower in $esr2a^{-/-}$ than in $esr2a^{+/+}$ males in the PPa, pPPp, and NVT (p=0.0359, 0.0430, and 0.0178, respectively) (*Figure 5C and D*), while no significant difference was observed in *arb* expression (*Figure 5—figure supplement 2C*). Considering that *ara* expression in these nuclei was also lower in $cyp19a1b^{-/-}$ males (*Figure 2F*), brain-derived estrogens presumably stimulate *ara* expression in these nuclei mainly through Esr2a. This idea was further supported by double-label in situ hybridization, which detected neurons coexpressing *ara* and *esr2a* in the PPa, pPPp, and NVT of wild-type males (*Figure 5—figure supplement 2D*). Examination of $esr2b^{+/+}$ and $esr2b^{-/-}$ male brains showed significantly lower expression of *ara* in the NAT and higher expression of *arb* in the PPa of $esr2b^{-/-}$ males (*Figure 5—figure supplement 2E–H*); however, no such changes in expression were observed in $cyp19a1b^{-/-}$ males (*Figure 2F and H*). It therefore seems unlikely that Esr2b mediates

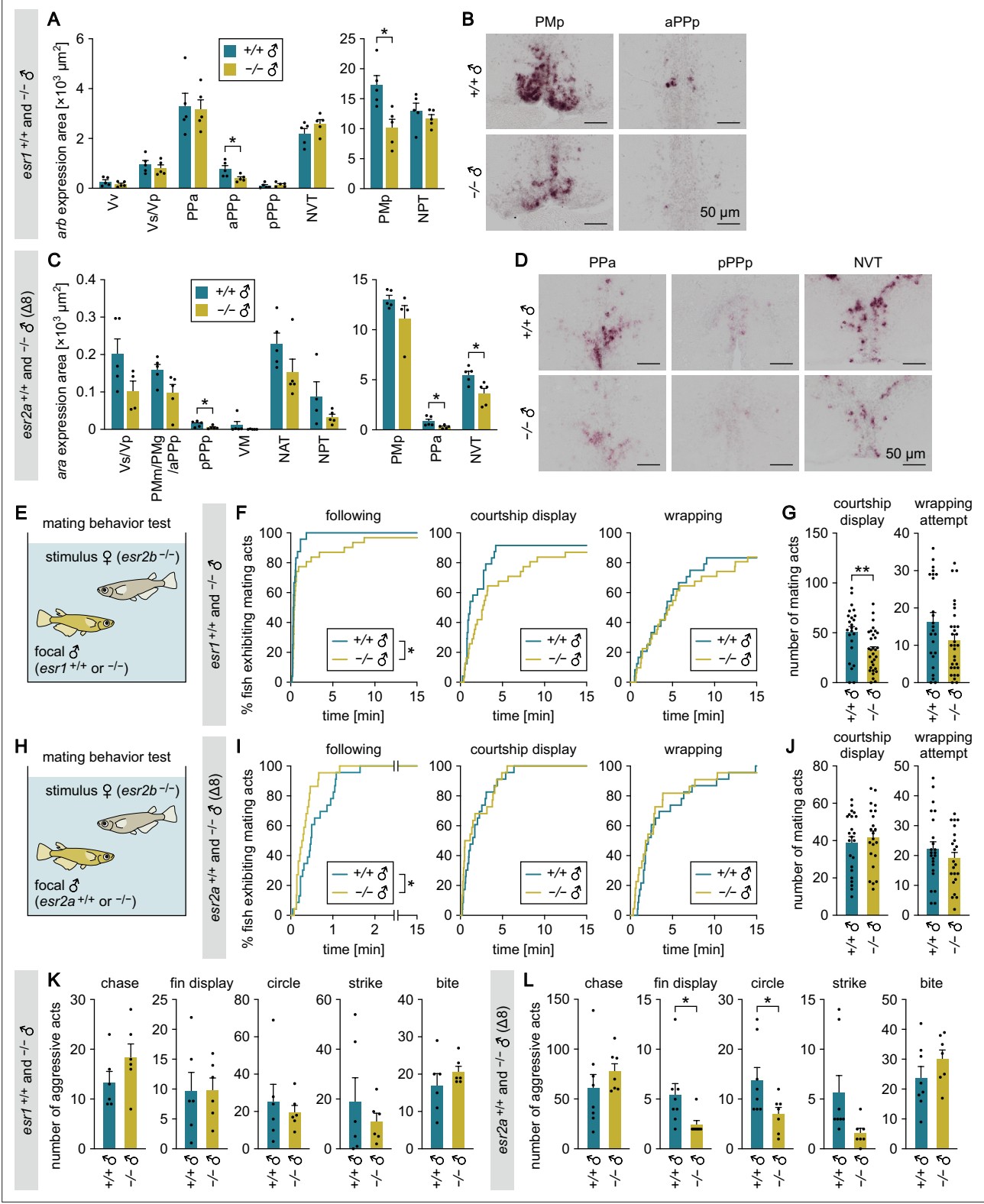

**Figure 5.** Brain-derived estrogens stimulate *ara* and *arb* expression in behaviorally relevant brain regions primarily through Esr2a and Esr1, respectively. (**A**) Total area of *arb* expression signal in each brain nucleus of *esr1*[+/+] and *esr1*[−/−] males (n=5 per genotype). The data are displayed in two graphs for visual clarity. (**B**) Representative images of *arb* expression in the PMp and aPPp. (**C**) Total area of *ara* expression signal in each brain nucleus of *esr2a*[+/+] and *esr2a*[−/−] males (Δ8 line; n=5 per genotype except for NPT of *esr2a*[+/+] males and Vs/Vp and PMp of *esr2a*[−/−] males, where n=4). The data are displayed in two graphs for visual clarity. (**D**) Representative images of *ara* expression in the PPa, pPPp, and NVT. (**E**) Setup for testing the mating

*Figure 5 continued on next page*

*Figure 5 continued*

behavior of *esr1*$^{+/+}$ and *esr1*$^{-/-}$ males using an *esr2b*-deficient female as the stimulus. (**F**) Latency of *esr1*$^{+/+}$ and *esr1*$^{-/-}$ males (n=24 and 31, respectively) to initiate each mating act toward the stimulus female. (**G**) Number of each mating act performed. (**H**) Setup for testing the mating behavior of *esr2a*$^{+/+}$ and *esr2a*$^{-/-}$ males using an *esr2b*-deficient female as the stimulus. (**I**) Latency of *esr2a*$^{+/+}$ and *esr2a*$^{-/-}$ males (Δ8 line; n=23 and 22, respectively) to initiate each mating act toward the stimulus female. (**J**) Number of each mating act performed. (**K, L**) Total number of each aggressive act observed among *esr1*$^{+/+}$ or *esr1*$^{-/-}$ males (n=6 per genotype) (**K**) and among *esr2a*$^{+/+}$ or *esr2a*$^{-/-}$ males (Δ8 line; n=8 and 7, respectively) (**L**) in the tank. Scale bars represent 50 μm. For abbreviations of brain nuclei, see ***Supplementary file 1***. Statistical differences were assessed by unpaired *t* test, with Welch's correction where appropriate (**A, C, G, J, K, L**) and Gehan-Breslow-Wilcoxon test (**F, I**). Error bars represent SEM. *p<0.05, **p<0.01.

The online version of this article includes the following source data and figure supplement(s) for figure 5:

**Source data 1.** Source data for *Figure 5*.

**Figure supplement 1.** Generation of *esr2a*-deficient medaka.

**Figure supplement 2.** Expression of *ara* and *arb* in the brain of males deficient for each ESR.

**Figure supplement 2—source data 1.** Source data for *Figure 5—figure supplement 2*.

**Figure supplement 3.** Mating behavior of *esr1*-deficient males.

**Figure supplement 3—source data 1.** Source data for *Figure 5—figure supplement 3*.

**Figure supplement 4.** Mating and aggressive behaviors of *esr2a*-deficient males.

**Figure supplement 4—source data 1.** Source data for *Figure 5—figure supplement 4*.

the effect of brain estrogens on AR expression. Collectively, these results suggest that brain-derived estrogens stimulate *ara* and *arb* expression in behaviorally relevant brain regions primarily through Esr2a and Esr1, respectively.

## *esr1*- and *esr2a*-deficient males are, respectively, less motivated to mate and less aggressive

If Esr1 and Esr2a truly mediate the behavioral effects of brain-derived estrogens through the activation of AR expression, then *esr1*- and *esr2a*-deficient males should exhibit impaired male-typical behaviors. More specifically, given that Esr1 and Esr2a function to activate the expression of *arb* and *ara*, which are responsible for male mating and aggression, respectively (***Nishiike and Okubo, 2024***), *esr1*- and *esr2a*-deficient males should be less motivated to mate and less aggressive, respectively.

To test these ideas, we first evaluated the mating behavior of *esr1*-deficient males using a wild-type female as the stimulus. As expected, *esr1*$^{-/-}$ males showed a significantly longer latency to initiate following than *esr1*$^{+/+}$ males (p=0.0055) (***Figure 5—figure supplement 3***), suggesting that they are less motivated to mate. This was further confirmed in tests using an *esr2b*-deficient female as the stimulus, where *esr1*$^{-/-}$ males showed a longer latency to following and fewer courtship displays than *esr1*$^{+/+}$ males (p=0.0426 and 0.0039, respectively) (***Figure 5E–G***). In contrast, no deficits were observed in the mating behavior of *esr2a*$^{-/-}$ males toward wild-type females (***Figure 5—figure supplement 4A and B***). Although *esr2a*$^{-/-}$ males of the Δ8 line showed a shorter latency to following than their wild-type siblings in tests using a stimulus *esr2b*-deficient female (p=0.0146) (***Figure 5H–J***), this was not reproduced in the Δ4 line (***Figure 5—figure supplement 4C and D***). We subsequently assessed aggressive behavior and found no defects in *esr1*$^{-/-}$ males (***Figure 5K***). Conversely, *esr2a*$^{-/-}$ males from both the Δ8 and Δ4 lines exhibited significantly fewer fin displays than their wild-type siblings (p=0.0461 and 0.0293, respectively). Circles followed a similar pattern, with a significant reduction in the Δ8 line (p=0.0446) and a comparable but nonsignificant decrease in the Δ4 line (p=0.1512) (***Figure 5L***, ***Figure 5—figure supplement 3E***), showing less aggression. Taken together with the previous finding that *esr2b*-deficient males show no deficits in either mating or aggressive behavior (***Nishiike et al., 2021***), these results suggest that brain-derived estrogens can promote mating with females by stimulating *arb* expression through Esr1 and can increase aggression toward other males by stimulating *ara* expression through Esr2a. Nonetheless, behavioral deficits in *esr1*- and *esr2a*-deficient males were relatively mild as compared with *cyp19a1b*-deficient males (***Figure 1***) and *ara*- and *arb*-deficient males (***Nishiike and Okubo, 2024***), suggesting that a compensatory mechanism may exist between ESR subtypes.

## Discussion

In this study, we found that male medaka deficient for *cyp19a1b* exhibit severely impaired male-typical mating and aggression. This observation was noteworthy because the fish had markedly elevated brain levels of 11KT, the primary driver of male-typical behaviors in teleosts (*Okubo et al., 2022*; *Kawabata-Sakata et al., 2024*). Deficits in mating were rescued by estrogen administration, indicating that reduced brain estrogen levels are the primary cause of the observed mating impairment; in other words, brain-derived estrogens are pivotal at least for male-typical mating behaviors in teleosts. This was also notable because, unlike in rodents, where brain-derived estrogens play essential roles in the establishment and activation of male-typical behaviors as posited by the 'aromatization hypothesis', these estrogens have been regarded as dispensable for male behaviors in many vertebrates, including teleosts (*Thornton et al., 2009*; *Bakker, 2022*; *Okubo et al., 2022*). In addition, the rescue of behavioral phenotypes by estrogen administration in adults suggests that in teleosts, unlike in rodents, brain-derived estrogens early in life are not essential for the execution of male-typical behaviors. While brain-derived estrogens are necessary for male behaviors in both rodents and teleosts, the life stages at which they exert their behavioral effects probably differ between these species. Brain aromatase activity in teleosts increases with age and, at adulthood, reaches 100–1000 times that in rodents (*Diotel et al., 2018*; *Okubo et al., 2022*). In contrast, brain aromatase activity in rodents reaches its peak during the perinatal period and thereafter declines with age (*Lephart, 1996*), although it remains important for male behavior in adulthood. This variation among species may represent the activation of brain estrogen synthesis at life stages critical for sexual differentiation of behavior that is unique to each species.

Brain-derived estrogens serve several functions during the process of sexual differentiation of the mouse brain, including the synthesis of prostaglandin PGE2 and the activation of synaptic and neurodevelopmental genes; however, the specific mechanism whereby these estrogens affect male behaviors remains obscure (*McCarthy et al., 2017*; *Gegenhuber et al., 2022*; *McCarthy, 2023*). Our findings in medaka indicate that brain-derived estrogens facilitate male-typical behaviors by potentiating androgen/AR signaling in behaviorally relevant brain regions via the direct stimulation of AR transcription. More specifically, they indicate that brain-derived estrogens (1) promote mating by stimulating the transcription of *arb* in some preoptic and hypothalamus nuclei via Esr1, and (2) increase aggression by stimulating the transcription of *ara* in other preoptic and hypothalamus nuclei through Esr2a. This model of the mechanism underlying the action of brain-derived estrogens would explain the apparent contradiction that, in teleosts, androgens per se elicit male behaviors without aromatization, while estrogen synthesis in the brain is also critical for these behaviors. Our data also indicate that the two AR genes, *ara* and *arb*, are direct downstream targets of brain-derived estrogens that mediate male behaviors, only a few of which have been identified thus far. In mice, perinatal brain estrogens increase *Ar* expression in the bed nucleus of the stria terminalis and preoptic area, two brain regions that have been implicated in male behaviors (*Juntti et al., 2010*). Recent evidence suggests that ESR1 binds to the regulatory genomic region of *Ar* in these brain regions in mice (*Gegenhuber et al., 2022*). Given these facts, the idea that brain-derived estrogens enhance androgen/AR signaling by directly stimulating AR transcription may apply to a wide range of species, including rodents.

Consistent with our results, studies in several teleost species have shown that treatment of males with an aromatase inhibitor reduces their male-typical behaviors, while estrogens exert the opposite effect (*Hallgren et al., 2006*; *O'Connell and Hofmann, 2012*; *Huffman et al., 2013*; *Jalabert et al., 2015*). Conversely, it has been shown in various teleosts, including medaka, that treatment with exogenous estrogens attenuates male behaviors (*Bayley et al., 1999*; *Bell, 2001*; *Bjerselius et al., 2001*; *Oshima et al., 2003*; *Martinović et al., 2007*). A possible explanation for this discrepancy is that estrogens may either stimulate or suppress male-typical behaviors, depending on their concentration. All studies showing the suppressive effects of exogenous estrogens were conducted at doses higher than those used in the present study or at doses mimicking the levels typical of adult females (*Kayo et al., 2020*). In addition, our previous study in male medaka showed that high doses of exogenous estrogens induce the expression of *esr2b*, which prevents male-typical mating behavior, in behaviorally relevant brain regions (*Nishiike et al., 2021*). Thus, the development of male behaviors may require moderate brain estrogen levels that are sufficient to induce the expression of *ara* and *arb*, but not *esr2b*, in the underlying neural circuitry. Considering this, the lack of aggression recovery in E2-treated *cyp19a1b*-deficient males in this study may be explained by the possibility that the E2 dose

used was sufficient to induce not only *ara* and *arb* but also *esr2b* expression in aggression-relevant circuits, which potentially suppressed aggression. Another possibility that is not mutually exclusive is that endogenous levels of brain estrogens are sufficient to motivate males to engage in male-typical behaviors, and therefore exogenous estrogens have no further effect. This possibility is at least likely for mating behavior, as estrogen treatment facilitated mating behavior in *cyp19a1b*-deficient males but not in their wild-type siblings. Further studies using *cyp19a1b* mutants from different teleost species are needed to explore these possibilities and to determine whether the findings in medaka hold for other teleosts.

In summary, we have shown that brain-derived estrogens promote male-typical behaviors by increasing brain sensitivity to testicular androgens through the stimulation of AR expression. Our findings challenge the prevailing view that brain-derived estrogens have little effect on male-typical behaviors in species where testicular androgens elicit these behaviors directly without aromatization to estrogens in the brain and unveil a previously unappreciated mechanism of the action of brain-derived estrogens. Because teleosts account for the majority of vertebrates, with rodents and some birds being the only known exceptions, the mechanism of their action that we have identified in medaka may be evolutionarily ancient and widely conserved across species.

# Materials and methods

**Key resources table**

| Reagent type (species) or resource | Designation | Source or reference | Identifiers | Additional information |
|---|---|---|---|---|
| Gene (*Oryzias latipes*) | *cyp19a1b* | GenBank | GenBank:AB591736 | |
| Gene (*O. latipes*) | *esr1* | GenBank | GenBank:XM_020714493 | |
| Gene (*O. latipes*) | *esr2a* | GenBank | GenBank:NM_001104702 | |
| Gene (*O. latipes*) | *esr2b* | GenBank | GenBank:XM_020713365 | |
| Gene (*O. latipes*) | *ara* | GenBank; NBRP Medaka | GenBank:NM_001122911; NBRP Medaka clone ID:olova36n18 | |
| Gene (*O. latipes*) | *arb* | GenBank | GenBank:NM_001104681 | |
| Gene (*O. latipes*) | *vt* | GenBank | GenBank:NM_001278891 | |
| Gene (*O. latipes*) | *gal* | GenBank | GenBank:LC532140 | |
| Gene (*O. latipes*) | *actb* | GenBank | GenBank:NM_001104808 | |
| Strain, strain background (*O. latipes*) | d-rR | NBRP Medaka | Strain ID:MT837 | Maintained in a closed colony over 15 years in Okubo lab |
| Genetic reagent (*O. latipes*) | *cyp19a1b*-deficient line | This paper | TILLING ID:57D05 | Generated and maintained in Okubo lab |
| Genetic reagent (*O. latipes*) | *esr1*-deficient line | https://doi.org/10.1093/pnasnexus/pgad413 | N/A | Generated and maintained in Okubo lab |
| Genetic reagent (*O. latipes*) | *esr2a*-deficient Δ8 line | This paper | N/A | Generated and maintained in Okubo lab |
| Genetic reagent (*O. latipes*) | *esr2a*-deficient Δ4 line | This paper | N/A | Generated and maintained in Okubo lab |

*Continued on next page*

*Continued*

| Reagent type (species) or resource | Designation | Source or reference | Identifiers | Additional information |
|---|---|---|---|---|
| Genetic reagent (*O. latipes*) | *esr2b*-deficient line | https://doi.org/10.1016/j.cub.2021.01.089 | TILLING ID:46E12 | Generated and maintained in Okubo lab |
| Cell line (*Homo sapiens*) | HEK293T | Riken BRC Cell Bank | cell number:RCB2202; RRID:CVCL_0063 | |
| Cell line (*H. sapiens*) | HeLa | Riken BRC Cell Bank | cell number:RCB0007; RRID:CVCL_0030 | |
| Antibody | Alkaline phosphatase-conjugated anti-DIG antibody (sheep polyclonal) | Roche Diagnostics | cat#:11093274910; RRID:AB_514497 | (1:500 or 1:2000) |
| Antibody | Anti-DIG antibody (mouse monoclonal) | Abcam | cat#:ab420; RRID:AB_304362 | (1:200) |
| Antibody | Horseradish peroxidase-conjugated anti-fluorescein antibody (sheep polyclonal) | PerkinElmer | cat#:NEF710001EA; RRID:AB_2737388 | (1:1000) |
| Recombinant DNA reagent | pcDNA3.1/V5-His-TOPO | Thermo Fisher Scientific | cat#:K480001 | |
| Recombinant DNA reagent | pGL4.10 | Promega | cat#:E6651 | |
| Recombinant DNA reagent | pGL4.74 | Promega | cat#:E6921 | |
| Recombinant DNA reagent | Medaka bacterial artificial chromosome (BAC) clone containing the *ara* locus | NBRP Medaka | NBRP Medaka clone ID:ola1-111G01 | |
| Recombinant DNA reagent | Medaka BAC clone containing the *arb* locus | NBRP Medaka | NBRP Medaka clone ID:ola1-192H15 | |
| Sequence-based reagent | CRISPR RNA (crRNA) for medaka *esr2a* | Fasmac | N/A | CTACGGCGTGTGGTCATGCGAGG |
| Sequence-based reagent | Trans-activating CRISPR RNA (tracrRNA) | Fasmac | cat#:GE-002 | |
| Peptide, recombinant protein | Cas9 | Nippon Gene | cat#:316-08651 | |
| Commercial assay or kit | Estradiol ELISA Kit | Cayman Chemical Company | cat#:582251 | |
| Commercial assay or kit | Testosterone ELISA Kit | Cayman Chemical Company | cat#:582701 | |
| Commercial assay or kit | 11-Keto Testosterone ELISA Kit | Cayman Chemical Company | cat#:582751 | |
| Commercial assay or kit | RNeasy Plus Universal Mini Kit | QIAGEN | cat#:73404 | |

*Continued*

| Reagent type (species) or resource | Designation | Source or reference | Identifiers | Additional information |
|---|---|---|---|---|
| Commercial assay or kit | SuperScript VILO cDNA Synthesis Kit | Thermo Fisher Scientific | cat#:11754050 | |
| Commercial assay or kit | LightCycler 480 SYBR Green I Master | Roche Diagnostics | cat#:04707516001 | |
| Commercial assay or kit | DIG RNA Labeling Mix | Roche Diagnostics | cat#:11277073910 | |
| Commercial assay or kit | Fluorescein RNA Labeling Mix | Roche Diagnostics | cat#:11685619910 | |
| Commercial assay or kit | T7 RNA polymerase | Roche Diagnostics | cat#:10881775001 | |
| Commercial assay or kit | SP6 RNA polymerase | Roche Diagnostics | cat#:10810274001 | |
| Commercial assay or kit | Dual-Luciferase Reporter Assay System | Promega | cat#:E1910 | |
| Commercial assay or kit | PrimeSTAR Mutagenesis Basal Kit | Takara Bio | cat#:R046A | |
| Commercial assay or kit | Alexa Fluor 555 Tyramide SuperBoost Kit, goat anti-mouse IgG | Thermo Fisher Scientific | cat#:B40913 | |
| Commercial assay or kit | TSA Plus Fluorescein System | PerkinElmer | cat#:NEL741001KT | |
| Commercial assay or kit | Sep-Pak C18 Plus Light Cartridge | Waters Corporation | cat#:WAT023501 | |
| Chemical compound, drug | Estradiol-17β (E2) | Fujifilm Wako Pure Chemical | cat#:058-04043 | |
| Chemical compound, drug | 5-Bromo-4-chloro-3-indolyl phosphate | Roche Diagnostics | cat#:11383221001 | |
| Chemical compound, drug | Nitro blue tetrazolium | Roche Diagnostics | cat#:11383213001 | |
| Chemical compound, drug | Lipofectamine LTX | Thermo Fisher Scientific | cat#:15338100 | |
| Chemical compound, drug | Charcoal/dextran-stripped fetal bovine serum | Cytiva | cat#:SH30068 | |
| Software, algorithm | AlphaFold 3 | https://alphafoldserver.com/about | RRID:SCR_025885 | |
| Software, algorithm | PyMOL | https://www.pymol.org | RRID:SCR_000305 | |
| Software, algorithm | Olyvia | Olympus | RRID:SCR_016167 | |
| Software, algorithm | Jaspar | http://jaspar.genereg.net/ | RRID:SCR_003030 | |

*Continued on next page*

*Continued*

| Reagent type (species) or resource | Designation | Source or reference | Identifiers | Additional information |
|---|---|---|---|---|
| Software, algorithm | Match | http://gene-regulation.com/pub/programs.html | RRID:SCR_007787 | |
| Software, algorithm | GraphPad Prism | GraphPad Software | RRID:SCR_002798 | |

## Animals and cell lines

Wild-type d-rR strain medaka and mutant medaka deficient for *cyp19a1b* and *esr2a* (generated in this study), *esr1* (*Fleming et al., 2023*), and *esr2b* (*Nishiike et al., 2021*) were maintained in a recirculating system at 28°C on a 14/10 hr light/dark cycle. Three or four times a day, they were fed live brine shrimp and dry food (Otohime; Marubeni Nisshin Feed, Tokyo, Japan). Sexually mature adults (2–6 months) were used for experiments and assigned randomly to experimental groups. Tissues were consistently sampled 1–5 hr after lights on. In each experiment, siblings raised under the same conditions were used as the comparison group to control for the effects of genetic diversity and environmental variation.

HEK293T and HeLa cells used in this study were confirmed to be mycoplasma-free (Biotherapy Institute of Japan, Tokyo, Japan) and authenticated by short tandem repeat profiling (National Institute of Biomedical Innovation, Osaka, Japan).

## Generation of mutant medaka

*cyp19a1b*-deficient medaka were generated essentially as previously described (*Nishiike et al., 2021*). In brief, a TILLING library of 5760 chemically mutagenized medaka (*Taniguchi et al., 2006*) was screened for mutations in exons 3, 4, and 5 of *cyp19a1b* by direct sequencing of PCR-amplified fragments. A founder (ID: 57D05) with a nonsense mutation in exon 4 (K105*) was identified (*Figure 1—figure supplement 1A and B*) and backcrossed to the wild-type d-rR strain for more than six generations to eliminate background mutations. Heterozygous males and females were intercrossed to generate wild-type, heterozygous, and homozygous siblings. All experimental fish were subjected to genotyping by PCR amplification across the mutation, followed by high-resolution melting (HRM) analysis, using the primers and probe listed in *Supplementary file 2*. HRM analysis was performed on the LightCycler 480 System II (Roche Diagnostics, Basel, Switzerland) using the LCGreen Plus dye (BioFire Defense, Salt Lake City, UT, USA).

*esr2a*-deficient medaka were generated by using the CRISPR/Cas9 system. A CRISPR RNA (crRNA) targeting exon 3 of *esr2a* (Fasmac, Kanagawa, Japan) (*Figure 5—figure supplement 1A*) was injected with trans-activating crRNA (Fasmac) and Cas9 protein (Nippon Gene Co. Ltd., Tokyo, Japan) into the cytoplasm of one- or two-cell stage embryos. The resulting fish were screened for germline mutations by outcrossing to wild-type fish and testing progeny for target site mutations by T7 endonuclease I assay (*Kim et al., 2009*) and direct sequencing. Two founders were identified that reproducibly yielded progeny carrying frameshift mutations that eliminated the DNA- and ligand-binding domains of Esr2a: one yielded progeny carrying an 8 bp deletion (Δ8); the other yielded progeny carrying a 4 bp deletion (Δ4) (*Figure 5—figure supplement 1A and B*). The DNA- and ligand-binding domains of medaka Esr2a were identified by sequence alignment with yellow perch (*Perca flavescens*) Esr2a, for which these domain locations have been reported (*Lynn et al., 2008*). The progeny were intercrossed to obtain wild-type, heterozygous, and homozygous siblings. The genotype of each experimental fish was determined by HRM analysis as described above.

A previous study reported that *esr2a*-deficient female medaka cannot release eggs due to oviduct atresia (*Kayo et al., 2019*). Likewise, some *esr2a*-deficient females generated in this study, despite the limited sample size, exhibited spawning behavior but were unable to release eggs (Δ8 line: 2/3; Δ4 line: 1/1), while such failure was not observed in wild-type females (Δ8 line: 0/12; Δ4 line: 0/11). These results support the effective loss of *esr2a* function.

## Measurement of sex steroid levels

Brain and peripheral levels of sex steroids were determined by enzyme-linked immunosorbent assay (ELISA) (*Nishiike et al., 2021*). Total lipids were extracted from the brain and from peripheral tissues, specifically the caudal half of the body excluding the head and visceral organs, of *cyp19a1b$^{+/+}$*, *cyp19a1b$^{+/–}$*, and *cyp19a1b$^{–/–}$* males with chloroform/methanol (2:1, vol/vol) and purified on a Sep-Pak C18 Plus Light Cartridge (Waters Corporation, Milford, MA, USA). Tissue levels of E2, testosterone, and 11KT were measured by using Estradiol, Testosterone, and 11-Keto Testosterone ELISA kits, respectively (Cayman Chemical Company, Ann Arbor, MI, USA).

## Real-time PCR

Total RNA was isolated from the brains of *cyp19a1b$^{+/+}$*, *cyp19a1b$^{+/–}$*, and *cyp19a1b$^{–/–}$* males using the RNeasy Plus Universal Mini Kit (QIAGEN, Hilden, Germany). cDNA was synthesized with the SuperScript VILO cDNA Synthesis Kit (Thermo Fisher Scientific, Waltham, MA, USA). Real-time PCR was performed on the LightCycler 480 System II using the LightCycler 480 SYBR Green I Master (Roche Diagnostics). Melting curve analysis was conducted to verify that a single amplicon was obtained in each sample. The β-actin gene (*actb*; GenBank accession number NM_001104808) was used to normalize the levels of target transcripts. The primers used for real-time PCR are shown in *Supplementary file 2*.

## Protein structure prediction

Structural predictions of Cyp19a1b proteins were conducted using AlphaFold 3 (*Abramson et al., 2024*). Amino acid sequences corresponding to the wild-type allele and the mutant allele generated in this study were submitted to the AlphaFold 3 prediction server. The resulting models were visualized with PyMOL (Schrödinger, New York, NY, USA), and key structural features, including the membrane helix, aromatic region, and heme-binding loop, were annotated.

## Mating behavior test

Mating behavior was tested essentially as previously described (*Hiraki-Kajiyama et al., 2019*). In brief, on the day before behavioral testing, each focal male (*cyp19a1b*-, *esr1*-, and *esr2a*-deficient lines) was placed with a stimulus female (wild-type females in the tests shown in *Figure 1D and E*, *Figure 5—figure supplement 3*, and *Figure 5—figure supplement 4A and B*; *esr2b*-deficient females in *Figures 1F–I*, *2A–E*, *5E–J* and *Figure 5—figure supplement 4C and D*) in a 2 l rectangular tank with a perforated transparent partition separating them. The setup was modified for E2-treated males, which were kept on E2 treatment and not introduced to the test tanks until the day of testing to ensure the efficacy of E2 treatment. The partition was removed 1 hr after lights on, and fish were allowed to interact for 30 min while their behavior was recorded with a digital video camera (HC-V360MS, HC-VX985M, or HC-W870M; Panasonic Corporation, Osaka, Japan). The first 15 min of the recording was used to calculate the latency to the first following, courtship display, wrapping, and spawning. In tests using an *esr2b*-deficient female as the stimulus fish, the latency to spawn could not be calculated because the female was unreceptive to males and did not spawn. Therefore, the sexual motivation of the focal male was assessed by counting the number of courtship displays and wrapping attempts in 30 min. To evaluate courtship displays performed by stimulus *esr2b*-deficient females toward focal males, the recording period was extended to 2 hr, as these females take longer to initiate courtship (*Nishiike et al., 2021*). In all video analyses, the researcher was blind to the fish genotype and treatment.

## Aggressive behavior test

To test male-male aggressive behavior (*Yamashita et al., 2020*), four males of the same genotype (*cyp19a1b*-, *esr1*-, and *esr2a*-deficient lines) that were unfamiliar with one another were placed together in a test tank 1 hr after lights on. After 1 min of acclimation to the tank, fish were allowed to interact for 15 min while their behavior was video-recorded as described above. The total number of each aggressive act (chase, fin display, circle, strike, and bite) displayed by the four males in the tank was counted manually from the video recordings.

## E2 treatment

*cyp19a1b$^{+/+}$* and *cyp19a1b$^{–/–}$* males were treated with 1 ng/ml of E2 (Fujifilm Wako Pure Chemical, Osaka, Japan), which was first dissolved in 100% ethanol (vehicle), or with the vehicle alone

by immersion in water for 4 days, with daily water changes to maintain the nominal concentration. The mating and aggressive behaviors of males before and after E2 treatment were evaluated as described above. The expression of *ara* and *arb* in the brain of E2- and vehicle-treated males was investigated by single-label in situ hybridization as described below. Although the exact increase in brain E2 levels following E2 treatment was not quantified, the observed positive effects on behavior and gene expression suggest that it was sufficient.

## Single-label in situ hybridization

Digoxigenin (DIG)-labeled cRNA probes were generated by in vitro transcription using DIG RNA Labeling Mix (Roche Diagnostics), T7 RNA polymerase (Roche Diagnostics), and the following cDNA fragments: *ara*, nucleotide 16–1030 of GenBank NM_001122911 (1015 bp); *arb*, 53–1233 of NM_001104681 (1181 bp); *vt*, 1–845 of NM_001278891 (845 bp); and *gal*, 5–533 of LC532140 (529 bp).

Single-label in situ hybridization was essentially done as described (*Kawabata-Sakata et al., 2020*). In brief, male brains from the *cyp19a1b*-deficient line (analysis of *ara*, *arb*, *vt*, and *gal*) and from the *esr1*-, *esr2a*-, and *esr2b*-deficient lines (analysis of *ara* and *arb*) were fixed in 4% paraformaldehyde and embedded in paraffin. Serial coronal sections of 10 µm thickness were hybridized with a DIG-labeled probe. Hybridization signal was detected with anti-DIG conjugated to alkaline phosphatase (RRID:AB_514497; Roche Diagnostics) and visualized using 5-bromo-4-chloro-3-indolyl phosphate/ nitro blue tetrazolium substrate (Roche Diagnostics). After color development for 15 min (*gal*), 2 hr (*vt*), or overnight (*ara* and *arb*), sections were imaged using a VS120 virtual slide microscope (Olympus, Tokyo, Japan). The total area of signal across all relevant sections, including both hemispheres, was calculated for each brain nucleus using Olyvia software (Olympus). Images were converted to a 256-level intensity scale, and pixels with intensities from 161 to 256 were considered signals. All sections used for comparison were processed in the same batch, without corrections between samples. Medaka brain atlases (*Anken and Bourrat, 1998*; *Ishikawa et al., 1999*) were used to identify brain nuclei.

## Transcriptional activity assay

Each full-length coding region of medaka Esr1 (GenBank XM_020714493), Esr2a (NM_001104702), and Esr2b (XM_020713365) cDNA was amplified by PCR and inserted into the expression vector pcDNA3.1/V5-His-TOPO (Thermo Fisher Scientific). Each nucleotide sequence of the 5′-flanking region of *ara* and *arb* was retrieved from the Ensembl medaka genome assembly and analyzed for potential canonical ERE-like sequences using Jaspar (version 5.0_alpha) and Match (public version 1.0) with default settings. No ERE-like sequence was detected in *ara*; therefore, the gene body and 3′-flanking region were also analyzed in this case. The transcription initiation site for *ara* and *arb* was determined based on an expressed sequence tag clone deposited in National BioResource Project (NBRP) Medaka (olova36n18) and GenBank NM_001104681, respectively. Medaka bacterial artificial chromosome (BAC) clones containing the *ara* (clone ID: ola1-111G01) and *arb* (ola1-192H15) loci were obtained from NBRP Medaka. A 4530 bp genomic DNA fragment upstream of *ara* exon 3 (comprising 2330 bp of the 5′-flanking region, exon 1, intron 1, exon 2, intron 2, and the first 32 bp of exon 3) was amplified by PCR from the BAC clone, fused to a P2A self-cleaving peptide sequence (*Kim et al., 2011*) at the 3′ end, and inserted into the NheI site of the luciferase reporter vector pGL4.10 (Promega, Madison, WI, USA). Similarly, a 3995 bp fragment upstream of the first methionine codon of *arb* (comprising 3815 bp of the 5′-flanking region and 180 bp of exon 1) was PCR-amplified and inserted into pGL4.10. The resulting constructs were transiently transfected into HEK293T (*ara*) or HeLa (*arb*) cells, together with an internal control vector pGL4.74 (Promega) and the Esr1, Esr2a, or Esr2b expression construct using Lipofectamine LTX (Thermo Fisher Scientific). Six hours after transfection, cells were incubated for 18 hr without and with E2 at doses of $10^{-11}$ M, $10^{-9}$ M, and $10^{-7}$ M in Dulbecco's modified Eagle's medium (phenol red-free) containing 5% charcoal/dextran-stripped fetal bovine serum (Cytiva, Marlborough, MA, USA). Luciferase activity was determined using the Dual-Luciferase Reporter Assay System (Promega) on the GloMax 20/20n Luminometer (Promega). All assays were conducted in duplicate or triplicate and repeated independently three times.

Further assays were performed with luciferase reporter constructs carrying point mutations in the ERE-like sequences to identify the ERE responsible for E2 induction of *ara* and *arb* transcription. Each half-site of the responsible ERE-like sequence was mutated into a HindIII recognition site using the

PrimeSTAR Mutagenesis Basal Kit (Takara Bio, Shiga, Japan). Transcriptional activity assays with these constructs were done as described above, except that a single dose of E2 ($10^{-7}$ M) was used.

## Double-label in situ hybridization

DIG-labeled cRNA probes for *ara* and *arb* were prepared as described above. Fluorescein-labeled cRNA probes for *esr1* and *esr2a* were generated by in vitro transcription using Fluorescein RNA Labeling Mix (Roche Diagnostics), SP6 or T7 RNA polymerase (Roche Diagnostics), and the following cDNA fragments: *esr1*, nucleotides 1694–2781 of XM_020714493 (1088 bp); *esr2a*, 1838–2276 of NM_001104702 plus 763 bp of 3'-untranslated sequence derived from the Ensembl medaka genome assembly (439 bp).

Double-label in situ hybridization was done as described previously (*Fleming et al., 2023*), with minor modifications. In brief, brains dissected from wild-type males were fixed in 4% paraformaldehyde, embedded in paraffin, and coronally sectioned at 10 μm thickness. Sections were hybridized simultaneously with the DIG-labeled *ara* or *arb* probe and fluorescein-labeled *esr1* or *esr2a* probe. DIG was detected with a mouse anti-DIG antibody (RRID:AB_304362; Abcam, Cambridge, UK) and visualized using the Alexa Fluor 555 Tyramide SuperBoost Kit, goat anti-mouse IgG (Thermo Fisher Scientific); fluorescein was detected with an anti-fluorescein antibody conjugated to horseradish peroxidase (RRID:AB_2737388; PerkinElmer, Waltham, MA, USA) and visualized using the TSA Plus Fluorescein System (PerkinElmer). Cell nuclei were counterstained with DAPI. Fluorescent images were acquired with a Leica TCS SP8 confocal laser scanning microscope (Leica Microsystems, Wetzlar, Germany) and the following excitation/emission wavelengths: 552/620–700 nm (Alexa Fluor 555), 488/495–545 nm (fluorescein), and 405/410–480 nm (DAPI). Cells were identified as coexpressing the two genes when Alexa Fluor 555 and fluorescein signals were clearly observed in the cytoplasm surrounding DAPI-stained nuclei, with intensities markedly stronger than the background noise.

## Statistical analysis

All quantitative data were expressed as mean ± standard error of the mean (SEM). On graphs, individual data points were plotted to indicate the underlying distribution. Behavioral time-series data were analyzed using Kaplan-Meier plots with the inclusion of fish that did not exhibit the given act within the test period.

Statistical analyses were done using GraphPad Prism (GraphPad Software, San Diego, CA, USA). Data between two groups were compared using unpaired two-tailed Student's *t* test. Welch's correction was applied if the *F* test indicated that the variance between groups was significantly different. The Bonferroni-Dunn correction was applied for multiple comparisons between two groups. To compare data among more than two groups, one-way analysis of variance (ANOVA) was performed, followed by either Bonferroni's (comparisons among experimental groups) or Dunnett's (comparisons between untreated and E2-treated groups in *Figure 4C and D*) post hoc test. If the Brown-Forsythe test indicated a significant difference in variance across groups, the data were instead analyzed using the non-parametric Kruskal-Wallis test, followed by Dunn's post hoc test. Differences between Kaplan-Meier curves were tested for significance using the Gehan-Breslow-Wilcoxon test (with Bonferroni's correction for more than two comparisons). Fish that spawned without any courtship display were excluded from the analysis of courtship display because it was not appropriate to treat them either as fish that did not perform courtship displays within the test duration or as fish that performed the first courtship display 0 s before spawning. Specifically, 7/18 *cyp19a1b*$^{+/+}$, 11/18 *cyp19a1b*$^{+/-}$, and 6/18 *cyp19a1b*$^{-/-}$ males were excluded in *Figure 1E*; 2/23 *esr1*$^{+/+}$ and 5/24 *esr1*$^{-/-}$ males were excluded in *Figure 5—figure supplement 3*; 2/24 *esr2a*$^{+/+}$ and 3/23 *esr2a*$^{-/-}$ males were excluded in *Figure 5—figure supplement 4A*; 0/23 *esr2a*$^{+/+}$ and 0/23 *esr2a*$^{-/-}$ males were excluded in *Figure 5—figure supplement 4B*. All data points were included in the analyses and no outliers were defined.

No power analysis was conducted due to the lack of relevant data; sample size was estimated based on previous studies reporting inter-individual variation in behavior and neural gene expression in medaka.

## Acknowledgements

We thank NBRP Medaka for providing the BAC clones, managing the TILLING library, and performing artificial insemination; Dr. Yoshihito Taniguchi for constructing the TILLING library; Dr. Masatoshi

Nakamoto for recovering the *cyp19a1b* mutant allele; Akira Hirata, Kaoru Furukawa, Tomiko Iba, and Ayu Kuwakubo for assistance with medaka husbandry.

## Additional information

### Competing interests

Kataaki Okubo: is an inventor on a patent application related to this work filed by Regional Fish Institute and the University of Tokyo (Japanese patent application no. JP2023-205555, filed on 5 December 2023). The other authors declare that no competing interests exist.

### Funding

| Funder | Grant reference number | Author |
| --- | --- | --- |
| Japan Society for the Promotion of Science | 21J20634 | Yuji Nishiike |
| Japan Society for the Promotion of Science | 23K26998 | Kataaki Okubo |
| Ministry of Education, Culture, Sports, Science and Technology | 17H06429 | Kataaki Okubo |
| Regional Fish Institute | Get-Research Grant | Kataaki Okubo |

The funders had no role in study design, data collection and interpretation, or the decision to submit the work for publication.

### Author contributions

Yuji Nishiike, Conceptualization, Formal analysis, Funding acquisition, Investigation, Writing – original draft; Shizuku Maki, Kaoru Ohno, Takeshi Usami, Formal analysis, Investigation, Writing – review and editing; Daichi Miyazoe, Conceptualization, Formal analysis, Investigation; Kiyoshi Nakasone, Formal analysis, Investigation; Yasuhiro Kamei, Takeshi Todo, Tomoko Ishikawa-Fujiwara, Investigation, Methodology, Writing – review and editing; Yoshitaka Nagahama, Conceptualization, Supervision, Writing – review and editing; Kataaki Okubo, Conceptualization, Supervision, Funding acquisition, Writing – original draft

### Author ORCIDs

Yuji Nishiike ![ORCID] https://orcid.org/0000-0001-6586-6348
Yasuhiro Kamei ![ORCID] https://orcid.org/0000-0001-6382-1365
Kataaki Okubo ![ORCID] https://orcid.org/0000-0002-4178-3094

### Ethics

All animal procedures were performed in accordance with the guidelines of the Institutional Animal Care and Use Committee of the University of Tokyo. The committee requests the submission of an animal-use protocol only for use of mammals, birds, and reptiles, in accordance with the Fundamental Guidelines for Proper Conduct of Animal Experiment and Related Activities in Academic Research Institutions under the jurisdiction of the Ministry of Education, Culture, Sports, Science and Technology of Japan (Ministry of Education, Culture, Sports, Science and Technology, Notice No. 71; June 1, 2006). Accordingly, we did not submit an animal-use protocol for this study, which used only teleost fish and thus did not require approval by the committee.

Reviewer #1 (Public review): https://doi.org/10.7554/eLife.97106.4.sa1
Reviewer #3 (Public review): https://doi.org/10.7554/eLife.97106.4.sa2
Author response https://doi.org/10.7554/eLife.97106.4.sa3

## Additional files

### Supplementary files
MDAR checklist

Supplementary file 1. Abbreviations of brain nuclei.

Supplementary file 2. Primers and probes used in this study.

### Data availability
All data supporting the findings of this study are included in the article and its supplementary information.

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
