## [Editor Report · eLife Assessment]

This is an overall **compelling** set of findings on the role of centrally produced estrogens in the control of behaviors in male medaka. The significance of the findings rests on the revealed potential mechanism between brain derived estrogens modulating social behaviors in males, supported by the analysis of multiple transgenic lines. The evidence for the broader claim is incomplete since it has not been extended to female medaka, and further experimentation would be necessary to fully validate the conclusions on the role of brain-derived estrogens. Nonetheless, the findings have led to **important** hypotheses on the hormonal control of behaviors in teleosts that can be tested further.

---

## [Referee Report · Reviewer #1 (Public review)]

Summary:

This research group has consistently performed cutting-edge research aiming to understand the role of hormones in the control of social behaviors, specifically by utilizing the genetically-tractable teleost fish, medaka, and the current work is no exception. The overall claim they make, that estrogens modulate social behaviors in males is supported, with important caveats. For one, there is no evidence these estrogens are generated by "neurons" as would be assumed by their main claim that it is NEUROestrogens that drive this effect. While indeed the aromatase they have investigated is expressed solely in the brain, in most teleosts, brain aromatase is only present in glial cells (astrocytes, radial glia). The authors should change this description so as not to mislead the reader. Below I detail more specific strengths and weaknesses of this manuscript.

Strengths:

Excellent use of the medaka model to disentangle the control of social behavior by sex steroid hormones

The findings are strong for the most part because deficits in the mutants are restored by the molecule (estrogens) that was no longer present due to the mutation

Presentation of the approach and findings are clear, allowing the reader to make their own inferences and compare them with the authors'

Includes multiple follow-up experiments, which leads to tests of internal replication and an impactful mechanistic proposal

Findings are provocative not just for teleost researchers, but for other species since, as the authors point out, the data suggest mechanisms of estrogenic control of social behaviors may be evolutionary ancient

Weaknesses:

The experimental design for studying aggression in males has flaws, but it appears a typical resident-intruder type assay is not appropriate for medaka. seems other species may be better for studying aggression in teleosts.

---

## [Referee Report · Reviewer #3 (Public review)]

Summary:

Taking advantage of the existence in fish of two genes coding for estrogen synthase, the enzyme aromatase, one mostly expressed in the brain (Cyp19a1b) and the other mostly found in the gonads (Cyp19a1a), this study investigates the role of brain-derived estrogens in the control of sexual and aggressive behavior in male medaka. The constitutive deletion of Cyp19a1b, confirmed by the ablation of its transcript, markedly reduced brain estrogen content. This effect is accompanied by reduced sexual and aggressive behavior and reduced expression of the transcripts coding for androgen receptors (AR), ara and arb, in brain regions involved in social behavior regulation. Both AR expression and aspects of social behaviors were restored by adult treatment with estrogens, providing some support for a role of aromatization. Expression analysis of AR isoforms and behavior of mutants of estrogen receptors (ER) indicates that these effects are likely mediated by the activation of the esr1 and esr2a isoforms. Together, these results provide valuable insights into the role of brain-derived estrogens in social behavior in fish.

Strengths:

This study evaluates the role of brain "specific" Cyp19a1 in the social behavior in male teleost fish, which as a taxon are more abundant and yet proportionally less studied that the most common birds and rodents. Therefore, evaluating the generalizability of results from higher vertebrates is important. The study suggests that, as opposed to mammals, the facilitatory role of brain-derived estrogens on mating and aggression is limited to adulthood.

Results obtained from multiple mutant lines converge to show that estrogens most likely synthesized in the brain drives aspects of male sexual behavior.

The comparative discussion of the age-dependent abundance of brain aromatase in fish vs mammals and its role in organization vs activation is important beyond the study of the targeted species.

Weaknesses:

Most experiments are weakly powered (low sample size).

The variability of the mRNA content for a same target gene between experiments (genotype comparison vs E2 treatment comparison) raises questions about the reproducibility of the data (apparent disappearance of genotype effect).

Conclusions :

Overall, the present study provides convincing evidence for a facilitatory role of estrogens originating from the brain on sexual behavior and aggressive behavior in male medaka. The role of specific estrogen receptor isoforms underlying the expression of androgen receptors is supported by converging evidence from multiple mutant lines.

---

## [Author Response]

The following is the authors’ response to the previous reviews.

**Public Reviews:**

**Reviewer #1 (Public Review):**
Summary:This research group has consistently performed cutting-edge research aiming to understand the role of hormones in the control of social behaviors, specifically by utilizing the genetically-tractable teleost fish, medaka, and the current work is no exception. The overall claim they make, that estrogens modulate social behaviors in males and females is supported, with important caveats. For one, there is no evidence these estrogens are generated by "neurons" as would be assumed by their main claim that it is NEUROestrogens that drive this effect. While indeed the aromatase they have investigated is expressed solely in the brain, in most teleosts, brain aromatase is only present in glial cells (astrocytes, radial glia). The authors should change this description so as not to mislead the reader. Below I detail more specific strengths and weaknesses of this manuscript.

We thank the reviewer for this positive evaluation of our work and for the helpful comments and suggestions. Regarding the concern that the term “neuroestrogens” may be misleading, we addressed this in the previous revision by consistently replacing it throughout the manuscript with “brain-derived estrogens” or “brain estrogens.”

In addition, the following sentence was added to the Introduction (line 61): “In teleost brains, including those of medaka, aromatase is exclusively localized in radial glial cells, in contrast to its neuronal localization in rodent brains (Forlano et al., 2001; Diotel et al., 2010; Takeuchi and Okubo, 2013).”

Strenghth:Excellent use of the medaka model to disentangle the control of social behavior by sex steroid hormonesThe findings are strong for the most part because deficits in the mutants are restored by the molecule (estrogens) that was no longer present due to the mutationPresentation of the approach and findings are clear, allowing the reader to make their own inferences and compare them with the authors'Includes multiple follow-up experiments, which leads to tests of internal replication and an impactful mechanistic proposalFindings are provocative not just for teleost researchers, but for other species since, as the authors point out, the data suggest mechanisms of estrogenic control of social behaviors may be evolutionary ancient

We thank the reviewer again for their positive evaluation of our work.

Weakness:As stated in the summary, the authors are attributing the estrogen source to neurons and there isn't evidence this is the case. The impact of the findings doesn't rest on this either

As mentioned above, we addressed this in the previous revision by replacing “neuroestrogens” with “brain-derived estrogens” or “brain estrogens” throughout the manuscript. In addition, the following sentence was added to the Introduction (line 61): “In teleost brains, including those of medaka, aromatase is exclusively localized in radial glial cells, in contrast to its neuronal localization in rodent brains (Forlano et al., 2001; Diotel et al., 2010; Takeuchi and Okubo, 2013).”

The d4 versus d8 esr2a mutants showed different results for aggression. The meaning and implications of this finding are not discussed, leaving the reader wondering

This comment is the same as one raised in the first review (Reviewer #1’s comment 2 on weaknesses), which we already addressed in our initial revision. For the reviewer’s convenience, we provide the response below:

Line 300: As the reviewer correctly noted, circles were significantly reduced in mutant males of the Δ8 line, whereas no significant reduction was observed in those of the Δ4 line. However, a tendency toward reduction was evident in the Δ4 line (P = 0.1512), and both lines showed significant differences in fin displays. Based on these findings, we believe our conclusion that esr2a^−/−^ males exhibit reduced aggression remains valid. To clarify this point and address potential reader concerns, we have revised the text as follows: “esr2a^−/−^ males exhibited significantly fewer fin displays (P = 0.0461 and 0.0293 for Δ8 and Δ4 lines, respectively) and circles (P = 0.0446 and 0.1512 for Δ8 and Δ4 lines, respectively) than their wild-type siblings (Fig. 5L; Fig. S8E), suggesting less aggression” was edited to read “esr2a^−/−^ males from both the Δ8 and Δ4 lines exhibited significantly fewer fin displays than their wild-type siblings (P = 0.0461 and 0.0293, respectively). Circles followed a similar pattern, with a significant reduction in the Δ8 line (P = 0.0446) and a comparable but non-significant decrease in the Δ4 line (P = 0.1512) (Figure 5L, Figure 5—figure supplement 3E), showing less aggression.”

Lack of attribution of previous published work from other research groups that would provide the proper context of the present study

This comment is also the same as one raised in the first review (Reviewer #1’s comment 3 on weaknesses). In our previous revision, in response to this comment, we cited the relevant references (Hallgren et al., 2006; O’Connell and Hofmann, 2012; Huffman et al., 2013; Jalabert et al., 2015; Yong et al., 2017; Alward et al., 2020; Ogino et al., 2023) in the appropriate sections. We also added the following new references and revised the Introduction and Discussion accordingly:

(2) Alward BA, Laud VA, Skalnik CJ, York RA, Juntti SA, Fernald RD. 2020. Modular genetic control of social status in a cichlid fish. Proceedings of the National Academy of Sciences of the United States of America 117:28167–28174. DOI: https://doi.org/10.1073/pnas.2008925117

(39) O’Connell LA, Hofmann HA. 2012. Social status predicts how sex steroid receptors regulate complex behavior across levels of biological organization. Endocrinology 153:1341–1351. DOI: https://doi.org/10.1210/en.2011-1663

(54) Yong L, Thet Z, Zhu Y. 2017. Genetic editing of the androgen receptor contributes to impaired male courtship behavior in zebrafish. Journal of Experimental Biology 220:3017–3021. DOI: https://doi.org/10.1242/jeb.161596

There are a surprising number of citations not included; some of the ones not included argue against the authors' claims that their findings were "contrary to expectation"

In our previous revision, we cited the relevant references (Hallgren et al., 2006; O’Connell and Hofmann, 2012; Huffman et al., 2013; Jalabert et al., 2015) in the Introduction. We also revised the text to remove phrases such as “contrary to expectation” and “unexpected.”

The experimental design for studying aggression in males has flaws. A standard test like a residentintruder test should be used.

Following this comment, we have attempted additional aggression assays using the resident-intruder paradigm. However, these experiments did not produce consistent or interpretable results. As noted in our previous revision, medaka naturally form shoals and exhibit weak territoriality, and even slight differences in dominance between a resident and an intruder can markedly increase variability, reducing data reliability. Therefore, we believe that the approach used in the present study provides a more suitable assessment of aggression in medaka, regardless of territorial tendencies. We will continue to explore potential refinements in future studies and respectfully ask the reviewer to evaluate the present work based on the assay used here.

While they investigate males and females, there are fewer experiments and explanations for the female results, making it feel like a small addition or an aside

While we did not adopt this comment in our previous revision, we have carefully reconsidered the reviewers’ feedback and have now decided to remove the female data. This change allows us to present a more focused and cohesive story centered on males. The specific revisions are outlined below:

Abstract

Line 25: The text “, thereby revealing a previously unappreciated mode of action of brain-derived estrogens. We additionally show that female fish lacking Cyp19a1b are less receptive to male courtship and conversely court other females, highlighting the significance of brain-derived estrogens in establishing sex-typical behaviors in both sexes.” has been revised to “. Taken together, these findings reveal a previously unappreciated mode of action of brain-derived estrogens in shaping male-typical behaviors.”

Results

Line 88: The text “Loss of cyp19a1b function in these fish was verified by measuring brain and peripheral levels of sex steroids. As expected, brain estradiol-17β (E2) in both male and female homozygous mutants (cyp19a1b^−/−^) was significantly reduced to 16% and 50%, respectively, of the levels in their wild-type (cyp19a1b^+/+^) siblings (P = 0.0037, males; P = 0.0092, females) (Fig. 1, A and B). In males, brain E2 in heterozygotes (cyp19a1b^−/−^) was also reduced to 45% of the level in wild-type siblings (P = 0.0284) (Fig. 1A), indicating a dosage effect of cyp19a1b mutation. In contrast, peripheral E2 levels were unaltered in both cyp19a1b^−/−^ males and females (Fig. S1, C and D), consistent with the expected functioning of Cyp19a1b primarily in the brain. Strikingly, brain levels of testosterone, as opposed to E2, increased 2.2-fold in cyp19a1b^−/−^ males relative to wild-type siblings (P = 0.0006) (Fig. 1A). Similarly, brain 11KT levels in cyp19a1b^−/−^ males and females increased 6.2- and 1.9-fold, respectively, versus wild-type siblings (P = 0.0007, males; P = 0.0316, females) (Fig. 1, A and B). These results show that cyp19a1b-deficient fish have reduced estrogen levels coupled with increased androgen levels in the brain, confirming the loss of cyp19a1b function. They also suggest that the majority of estrogens in the male brain and half of those in the female brain are synthesized locally in the brain. In addition, peripheral 11KT levels in cyp19a1b^−/−^ males and females increased 3.7- and 1.8-fold, respectively (P = 0.0789, males; P = 0.0118, females) (Fig. S1, C and D), indicating peripheral influence in addition to central effects.” has been revised to “Loss of cyp19a1b function in these fish was verified by measuring brain and peripheral levels of sex steroids in males. As expected, brain estradiol-17β (E2) in homozygous mutants (cyp19a1b^−/−^) was significantly reduced to 16% of the levels in wild-type (cyp19a1b^+/+^) siblings (P = 0.0037) (Figure 1A). Brain E2 in heterozygotes (cyp19a1b^+/−^) was also reduced to 45% of wild-type levels (P = 0.0284) (Figure 1A), indicating a dosage effect of the cyp19a1b mutation. In contrast, peripheral E2 levels were unaltered in cyp19a1b^−/−^ males (Figure 1B), consistent with the expected functioning of Cyp19a1b primarily in the brain. Strikingly, brain testosterone levels, as opposed to E2, increased 2.2-fold in cyp19a1b^−/−^ males relative to wild-type siblings (P = 0.0006) (Figure 1A). Similarly, brain 11KT levels increased 6.2-fold (P = 0.0007) (Figure 1A). These results indicate that cyp19a1b-deficient males have reduced estrogen coupled with elevated androgen levels in the brain, confirming the loss of cyp19a1b function. They also suggest that the majority of estrogens in the male brain are synthesized locally in the brain. Peripheral 11KT levels also increased 3.7-fold in cyp19a1b^−/−^ males (P = 0.0789) (Figure 1B), indicating peripheral influence in addition to central effects.”

Line 211: “expression of vt in the pNVT of cyp19a1b^−/−^ males was significantly reduced to 18% as compared with cyp19a1b^+/+^ males (P = 0.0040), a level comparable to that observed in females” has been revised to “expression of vt in the pNVT of cyp19a1b^−/−^ males was significantly reduced to 18% as compared with cyp19a1b^+/+^ males (P = 0.0040).”

The subsection entitled “cyp19a1b-deficient females are less receptive to males and instead court other females,” which followed line 311, has been removed.

Discussion

The two paragraphs between lines 373 and 374, which addressed the female data, have been removed.

Materials and methods

Line 433: “males and females” has been changed to “males”.

Line 457: “focal fish” has been changed to “focal male”.

Line 458: “stimulus fish” has been changed to “stimulus female”.

Line 458: “Fig. 6, E and F, ” has been deleted.

Line 460: “; wild-type males in Fig. 6, A to C” has been deleted.

Line 466: The text “The period of interaction/recording was extended to 2 hours in tests of courtship displays received from the stimulus esr2b-deficient female and in tests of mating behavior between females, because they take longer to initiate courtship (12). In tests using an esr2b-deficient female as the stimulus fish, where the latency to spawn could not be calculated because these fish were unreceptive to males and did not spawn, the sexual motivation of the focal fish was instead assessed by counting the number of courtship displays and wrapping attempts in 30 min. The number of these mating acts was also counted in tests to evaluate the receptivity of females. In tests of mating behavior between two females, the stimulus female was marked with a small notch in the caudal fin to distinguish it from the focal female.” has been revised to “In tests using an esr2b-deficient female as the stimulus fish, the latency to spawn could not be calculated because the female was unreceptive to males and did not spawn. Therefore, the sexual motivation of the focal male was assessed by counting the number of courtship displays and wrapping attempts in 30 min. To evaluate courtship displays performed by stimulus esr2bdeficient females toward focal males, the recording period was extended to 2 hours, as these females take longer to initiate courtship (Nishiike et al., 2021). In all video analyses, the researcher was blind to the fish genotype and treatment.”

Line 499: “brains dissected from males and females of the cyp19a1b-deficient line (analysis of ara, arb, vt, gal, npba, and esr2b) and males of the esr1-, esr2a-, and esr2b-deficient lines” has been revised to “male brains from the cyp19a1b-deficient line (analysis of ara, arb, vt, and gal) and from the esr1-, esr2a-, and esr2b-deficient lines.”

Line 504: “After color development for 15 min (gal), 40 min (npba), 2 hours (vt), or overnight (ara, arb, and esr2b)” has been revised to “After color development for 15 min (gal), 2 hours (vt), or overnight (ara and arb).”

Line 516: “Thermo Fisher Scientific, Waltham, MA” has been changed to “Thermo Fisher Scientific” to avoid redundancy.

Line 565: The subsection entitled “Measurement of spatial distances between fish” has been removed.

Line 585: “6/10 cyp19a1b^+/+^, 3/10 cyp19a1b^+/−^, and 6/10 cyp19a1b^−/−^ females were excluded in Fig. 6B;” has been deleted.

References

The following references have been removed:

Capel B. 2017. Vertebrate sex determination: evolutionary plasticity of a fundamental switch. Nature Reviews Genetics 18:675–689. DOI: https://doi.org/10.1038/nrg.2017.60

Hiraki T, Nakasone K, Hosono K, Kawabata Y, Nagahama Y, Okubo K. 2014. Neuropeptide B is femalespecifically expressed in the telencephalic and preoptic nuclei of the medaka brain. Endocrinology 155:1021–1032. DOI: https://doi.org/10.1210/en.2013-1806

Juntti SA, Hilliard AT, Kent KR, Kumar A, Nguyen A, Jimenez MA, Loveland JL, Mourrain P, Fernald RD. 2016. A neural basis for control of cichlid female reproductive behavior by prostaglandin F2α. Current Biology 26:943–949. DOI: https://doi.org/10.1016/j.cub.2016.01.067

Kimchi T, Xu J, Dulac C. 2007. A functional circuit underlying male sexual behaviour in the female mouse brain. Nature 448:1009–1014. DOI: https://doi.org/10.1038/nature06089

Kobayashi M, Stacey N. 1993. Prostaglandin-induced female spawning behavior in goldfish (Carassius auratus) appears independent of ovarian influence. Hormones and Behavior 27:38–55. DOI: https://doi.org/10.1006/hbeh.1993.1004

Liu H, Todd EV, Lokman PM, Lamm MS, Godwin JR, Gemmell NJ. 2017. Sexual plasticity: a fishy tale. Molecular Reproduction and Development 84:171–194. DOI: https://doi.org/10.1002/mrd.22691

Munakata A, Kobayashi M. 2010. Endocrine control of sexual behavior in teleost fish. General and Comparative Endocrinology 165:456–468. DOI: https://doi.org/10.1016/j.ygcen.2009.04.011

Nugent BM, Wright CL, Shetty AC, Hodes GE, Lenz KM, Mahurkar A, Russo SJ, Devine SE, McCarthy MM. 2015. Brain feminization requires active repression of masculinization via DNA methylation. Nature Neuroscience 18:690–697. DOI: https://doi.org/10.1038/nn.3988

Shaw K, Therrien M, Lu C, Liu X, Trudeau VL. 2023. Mutation of brain aromatase disrupts spawning behavior and reproductive health in female zebrafish. Frontiers in Endocrinology 14:1225199. DOI: https://doi.org/10.3389/fendo.2023.1225199

Stacey NE. 1976. Effects of indomethacin and prostaglandins on the spawning behaviour of female goldfish. Prostaglandins 12:113–126. DOI: https://doi.org/10.1016/s0090-6980(76)80010-x

Figure 1

Panel B, which originally showed steroid levels in female brains, has been replaced with steroid levels in the periphery of males, originally presented in Figure S1, panel C. Accordingly, the legend “(A and B) Levels of E2, testosterone, and 11KT in the brain of adult cyp19a1b^+/+^, cyp19a1b^+/−^, and cyp19a1b^−/−^ males (A) and females (B) (n = 3 per genotype and sex).” has been revised to “(A, B) Levels of E2, testosterone, and 11KT in the brain (A) and periphery (B) of adult cyp19a1b^+/+^, cyp19a1b^+/−^, and cyp19a1b^−/−^ males (n = 3 per genotype).”

Figure 3

The female data have been deleted from Figure 3. The revised Figure 3 is presented.

The corresponding legend text has been revised as follows:

Line 862: “males and females (n = 4 and 5 per genotype for males and females, respectively)” has been changed to “males (n = 4 per genotype)”.

Line 864: “males and females (n = 4 except for cyp19a1b^+/+^ males, where n = 3)” has been changed to “males (n = 3 and 4, respectively)”.

Figure 6

Figure 6 and its legend have been removed.

Figure 1—figure supplement 1

Panel C, showing male data, has been moved to Figure 1B, as described above, while panel D, showing female data, has been deleted. The corresponding legend “(**C** and **D**) Levels of E2, testosterone, and 11KT in the periphery of adult cyp19a1b^+/+^, cyp19a1b^+/−^, and cyp19a1b^−/−^ males (C) and females (D) (n = 3 per genotype and sex). Statistical differences were assessed by Bonferroni’s post hoc test (C and D). Error bars represent SEM. *P < 0.05.” has also been removed.

Line 804: Following this change, the figure title has been updated from “Generation of cyp19a1bdeficient medaka and evaluation of peripheral sex steroid levels” to “Generation of cyp19a1b-deficient medaka.”

The statistics comparing "experimental to experimental" and "control to experimental" isn't appropriate

This comment is the same as one raised in the first review (Reviewer #1’s comment 7 on weaknesses), which we already addressed in our initial revision. For the reviewer’s convenience, we provide the response below:

The reviewer raised concerns about the statistical analysis used for Figures 4C and 4E, suggesting that Bonferroni’s test should be used instead of Dunnett’s test. However, Dunnett’s test is commonly used to compare treatment groups to a reference group that receives no treatment, as in our study. Since we do not compare the treated groups with each other, we believe Dunnett’s test is the most appropriate choice.

Line 576: The reviewer’s concern may have arisen from the phrase “comparisons between control and experimental groups” in the Materials and methods. We have revised it to “comparisons between untreated and E2-treated groups in Figure 4C and D” for clarity.

**Reviewer #3 (Public Review):**
Summary:Taking advantage of the existence in fish of two genes coding for estrogen synthase, the enzyme aromatase, one mostly expressed in the brain (Cyp19a1b) and the other mostly found in the gonads (Cyp19a1a), this study investigates the role of brain-derived estrogens in the control of sexual and aggressive behavior in medaka. The constitutive deletion of Cyp19a1b markedly reduced brain estrogen content in males and to a lesser extent in females. These effects are accompanied by reduced sexual and aggressive behavior in males and reduced preference for males in females. These effects are reversed by adult treatment with supporting a role for estrogens. The deletion of Cyp19a1b is associated with a reduced expression of the genes coding for the two androgen receptors, ara and arb, in brain regions involved in the regulation of social behavior. The analysis of the gene expression and behavior of mutants of estrogen receptors indicates that these effects are likely mediated by the activation of the esr1 and esr2a isoforms. These results provide valuable insight into the role of estrogens in social behavior in the most abundant vertebrate taxon, however the conclusion of brain-derived estrogens awaits definitive confirmation.

We thank this reviewer for their positive evaluation of our work and comments that have improved the manuscript.

Strength:Evaluation of the role of brain "specific" Cyp19a1 in male teleost fish, which as a taxon are more abundant and yet proportionally less studied that the most common birds and rodents. Therefore, evaluating the generalizability of results from higher vertebrates is important. This approach also offers great potential to study the role of brain estrogen production in females, an understudied question in all taxa.Results obtained from multiple mutant lines converge to show that estrogen signaling, likely synthesized in the brain drives aspects of male sexual behavior.The comparative discussion of the age-dependent abundance of brain aromatase in fish vs mammals and its role in organization vs activation is important beyond the study of the targeted species. - The authors have made important corrections to tone down some of the conclusions which are more in line with the results.

We thank the reviewer again for their positive evaluation of our work and the revisions we have made.

weaknesses:No evaluation of the mRNA and protein products of Cyp19a1b and ESR2a are presented, such that there is no proper demonstration that the mutation indeed leads to aromatase reduction. The conclusion that these effects dependent on brain derived estrogens is therefore only supported by measures of E2 with an EIA kit that is not validated. No discussion of these shortcomings is provided in the discussion thus further weakening the conclusion manuscript.

In response to this and other comments, we have now provided direct validation that the cyp19a1b mutation in our medaka leads to loss of function. Real-time PCR analysis showed that cyp19a1b transcript levels in the brain were reduced by approximately half in cyp19a1b^+/−^ males and were nearly absent in cyp19a1b^−/−^ males, consistent with nonsense-mediated mRNA decay

In addition, AlphaFold 3-based structural modeling indicated that the mutant Cyp19a1b protein lacks essential motifs, including the aromatic region and heme-binding loop, and exhibits severe conformational distortion (see figure; key structural features are annotated as follows: membrane helix (blue), aromatic region (red), and heme-binding loop (orange)).

Results:

Line 101: The following text has been added: “Loss of cyp19a1b function was further confirmed by measuring cyp19a1b transcript levels in the brain and by predicting the three-dimensional structure of the mutant protein. Real-time PCR revealed that transcript levels were reduced by half in cyp19a1b^+/−^ males and were nearly undetectable in cyp19a1b^−/−^ males, presumably as a result of nonsense-mediated mRNA decay (Lindeboom et al., 2019) (Figure 1C). The wild-type protein, modeled by AlphaFold 3, exhibited a typical cytochrome P450 fold, including the membrane helix, aromatic region, and hemebinding loop, all arranged in the expected configuration (Figure 1—figure supplement 1C). The mutant protein, in contrast, was severely truncated, retaining only the membrane helix (Figure 1—figure supplement 1C). The absence of essential domains strongly indicates that the allele encodes a nonfunctional Cyp19a1b protein. Together, transcript and structural analyses consistently demonstrate that the mutation generated in this study causes a complete loss of cyp19a1b function.”

Materials and methods

Line 438: A subsection entitled “Real-time PCR” has been added. The text of this subsection is as follows: “Total RNA was isolated from the brains of cyp19a1b^+/+^, cyp19a1b^+/−^, and cyp19a1b^−/−^ males using the RNeasy Plus Universal Mini Kit (Qiagen, Hilden, Germany). cDNA was synthesized with the SuperScript VILO cDNA Synthesis Kit (Thermo Fisher Scientific, Waltham, MA). Real-time PCR was performed on the LightCycler 480 System II using the LightCycler 480 SYBR Green I Master (Roche Diagnostics). Melting curve analysis was conducted to verify that a single amplicon was obtained in each sample. The β-actin gene (actb; GenBank accession number NM_001104808) was used to normalize the levels of target transcripts. The primers used for real-time PCR are shown in Supplementary file 2.”

Line 448: A subsection entitled “Protein structure prediction” has been added. The text of this subsection is as follows: “Structural predictions of Cyp19a1b proteins were conducted using AlphaFold 3 (Abramson et al., 2024). Amino acid sequences corresponding to the wild-type allele and the mutant allele generated in this study were submitted to the AlphaFold 3 prediction server. The resulting models were visualized with PyMOL (Schrödinger, New York, NY), and key structural features, including the membrane helix, aromatic region, and heme-binding loop, were annotated.”

References

The following two references have been added:

Abramson J, Adler J, Dunger J, Evans R, Green T, Pritzel A, Ronneberger O, Willmore L, Ballard AJ, Bambrick J, Bodenstein SW, Evans DA, Hung CC, O'Neill M, Reiman D, Tunyasuvunakool K, Wu Z, Žemgulytė A, Arvaniti E, Beattie C, Bertolli O, Bridgland A, Cherepanov A, Congreve M, CowenRivers AI, Cowie A, Figurnov M, Fuchs FB, Gladman H, Jain R, Khan YA, Low CMR, Perlin K, Potapenko A, Savy P, Singh S, Stecula A, Thillaisundaram A, Tong C, Yakneen S, Zhong ED, Zielinski M, Žídek A, Bapst V, Kohli P, Jaderberg M, Hassabis D, Jumper JM. 2024. Accurate structure prediction of biomolecular interactions with AlphaFold 3. Nature **630**:493–500. DOI: https://doi.org/10.1038/s41586-024-07487-w

Lindeboom RGH, Vermeulen M, Lehner B, Supek F. 2019. The impact of nonsense-mediated mRNA decay on genetic disease, gene editing and cancer immunotherapy. Nature Genetics **51**:1645–1651. DOI: https://doi.org/10.1038/s41588-019-0517-5

Figure 1

The real-time PCR results described above have been incorporated in Figure 1, panel C, with the corresponding legend provided below (line 788).

(C) Brain cyp19a1b transcript levels in cyp19a1b^+/+^, cyp19a1b^+/−^, and cyp19a1b^−/−^ males (n = 6 per genotype). Mean value for cyp19a1b^+/+^ males was arbitrarily set to 1.

The subsequent panels have been renumbered accordingly. The entirety of the revised Figure 1.

Figure 1—figure supplement 1

The AlphaFold 3-generated structural models described above have been incorporated in Figure 1— figure supplement 1, panel C, with the corresponding legend provided below (line 811).

(C) Predicted three-dimensional structures of wild-type (left) and mutant (right) Cyp19a1b proteins. Key structural features are annotated as follows: membrane helix (blue), aromatic region (red), and heme-binding loop (orange).

The entirety of the revised Figure 1—figure supplement 1 is presented

The information on the primers used for real-time PCR has been included in Supplementary file 2.

The functional deficiency of esr2a was already addressed in the previous revision. For clarity, we have reproduced the relevant information here.

A previous study reported that female medaka lacking esr2a fail to release eggs due to oviduct atresia (Kayo et al., 2019, Sci Rep 9:8868). Similarly, in this study, some esr2a-deficient females exhibited spawning behavior but were unable to release eggs, although the sample size was limited (Δ8 line: 2/3; Δ4 line: 1/1). In contrast, this was not observed in wild-type females (Δ8 line: 0/12; Δ4 line: 0/11). These results support the effective loss of esr2a function. To incorporate this information into the manuscript, the following text has been added to the Materials and methods (line 423): “A previous study reported that esr2a-deficient female medaka cannot release eggs due to oviduct atresia (Kayo et al., 2019). Likewise, some esr2a-deficient females generated in this study, despite the limited sample size, exhibited spawning behavior but were unable to release eggs (Δ8 line: 2/3; Δ4 line: 1/1), while such failure was not observed in wild-type females (Δ8 line: 0/12; Δ4 line: 0/11). These results support the effective loss of esr2a function.”

Most experiments are weakly powered (low sample size).

This comment is essentially the same as one raised in the first review (Reviewer #3’s comment 7 on weaknesses). We acknowledge the reviewer’s concern that the histological analyses were weakly powered due to the limited sample size. In our earlier revision, we responded as follows:

Histological analyses were conducted with a relatively small sample size, as our previous experience suggested that interindividual variability in the results would not be substantial. Since significant differences were detected in many analyses, further increasing the sample size was deemed unnecessary.

The variability of the mRNA content for a same target gene between experiments (genotype comparison vs E2 treatment comparison) raises questions about the reproducibility of the data (apparent disappearance of genotype effect).

This comment is the same as one raised in the first review (Reviewer #3’s comment 8 on weaknesses), which we already addressed in our initial revision. For the reviewer’s convenience, we provide the response below:

As the reviewer pointed out, the overall area of ara expression is larger in Figure 2J than in Figure 2F. However, the relative area ratios of ara expression among brain nuclei are consistent between the two figures, indicating the reproducibility of the results. Thus, this difference is unlikely to affect the conclusions of this study.

Additionally, the differences in ara expression in pPPp and arb expression in aPPp between wild-type and cyp19a1b-deficient males appear less pronounced in Figures 2J and 2K than in Figures 2F and 2H. This is likely attributable to the smaller sample size used in the experiments for Figures 2J and 2K, resulting in less distinct differences. However, as the same genotype-dependent trends are observed in both sets of figures, the conclusion that ara and arb expression is reduced in cyp19a1b-deficient male brains remains valid.

Conclusions:Overall, the claims regarding role of estrogens originating in the brain on male sexual behavior is supported by converging evidence from multiple mutant lines. The role of brain-derived estrogens on gene expression in the brain is weaker as are the results in females.

We appreciate the reviewer’s positive evaluation of our findings on male behavior. The concern regarding the role of brain-derived estrogens in gene expression has been addressed in our rebuttal, and the female data have been removed so that the analysis now focuses on males. The specific revisions for removing the female data are described in Response to reviewer #1’s comment 6 on weaknesses.

**Recommendations For The Authors:**

**Reviewer #1 (Recommendations For The Authors):**
The manuscript is improved slightly. I am thankful the authors addressed some concerns, but for several concerns the referees raised, the authors acknowledged them yet did not make corresponding changes to the manuscript or disagreed that they were issues at all without explanation. All reviewers had issues with the imbalanced focus on males versus females and the male aggression assay. Yet, they did not perform additional experiments or even make changes to the framing and scope of the manuscript. If the authors had removed the female data, they may have had a more cohesive story, but then they would still be left with inadequate behavior assays in the males. If the authors don't have the time or resources to perform the additional work, then they should have said so. However, the work would be incomplete relative to the claims. That is a key point here. If they change their scope and claims, the authors avoid overstating their findings. I want to see this work published because I believe it moves the field forward. But the authors need to be realistic in their interpretations of their data.

In response to this and related comments, we have removed the female data and focused the manuscript on analyses in males. The specific revisions are described in Response to reviewer #1’s comment 6 on weaknesses. Additionally, we have validated that the cyp19a1b mutation in our medaka leads to loss of function (see Response to reviewer #3’s comment 1 on weaknesses), which further strengthens the reliability of our conclusions regarding male behavior.

I agree with the reviewer who said we need to see validation of the absence of functional cyp19a1 b in the brain. However, the results from staining for the protein and performing in situ could be quizzical. Indeed, there aren't antibodies that could distinguish between aromatase a and b, and it is not uncommon for expression of a mutated gene to be normal. One approach they could do is measure aromatase activity, but they are *sort of* doing that by measuring brain E2. It's not perfect, but we teleost folks are limited in these areas. At the very least, they should show the predicted protein structure of the mutated aromatase alleles. It could show clearly that the tertiary structure is utterly absent, giving more support to the fact that their aromatase gene is non-functional.

As noted above, we have further validated the loss of cyp19a1b function by measuring cyp19a1b transcript levels in the brain and predicting the three-dimensional structure of the mutant protein. These analyses confirmed that cyp19a1b function is indeed lost, thereby increasing the reliability of our conclusions. For further details, please refer to Response to reviewer #3’s comment 1 on weaknesses.

With all of this said, the work is important, and it is possible that with a reframing of the impact of their work in the context of their findings, I could consider the work complete. I think with a proper reframing, the work is still impactful.

In accordance with this feedback, and as described above, we have reframed the manuscript by removing the female data and focusing exclusively on males. This revision clarifies the scope of our study and reinforces the support for our conclusions. For further details, please refer to Response to reviewer #1’s comment 6 on weaknesses.

(1) Clearly state in the Figure 1 legend that each data point for male aggressive behaviors represents the total # of behaviors calculated over the 4 males in each experimental tank.

In response to this comment, we have revised the legend of Figure 1K (line 797). The original legend, “(K) Total number of each aggressive act observed among cyp19a1b^+/+^, cyp19a1b^+/−^, or cyp19a1^−/−^ males in the tank (n = 6, 7, and 5, respectively),” has been updated to “(K) Total number of each aggressive act performed by cyp19a1b^+/+^, cyp19a1b^+/−^, and cyp19a1b^−/−^ males. Each data point represents the sum of acts recorded for the 4 males of the same genotype in a single tank (n = 6, 7, and 5 tanks, respectively).” This clarifies that each data point reflects the total behaviors of the 4 males within each tank.

(2) The authors wrote under "Response to reviewer #1's major comment "...the development of male behaviors may require moderate neuroestrogen levels that are sufficient to induce the expression of ara and arb, but not esr2b, in the underlying neural circuitry": "This may account for the lack of aggression recovery in E2-treated cyp19a1b-deficient males in this study.".What is meant by the latter statement? What accounts for the lack of aggression? The lack of increase in esr2b? Please clarify.

Line 365: In response to this comment, “This may account for the lack of aggression recovery in E2treated cyp19a1b-deficient males in this study.” has been revised to “Considering this, the lack of aggression recovery in E2-treated cyp19a1b-deficient males in this study may be explained by the possibility that the E2 dose used was sufficient to induce not only ara and arb but also esr2b expression in aggression-relevant circuits, which potentially suppressed aggression.”

This revision clarifies that, while moderate brain estrogen levels are sufficient to promote male behaviors via induction of ara and arb, the E2 dose used in this study may have additionally induced esr2b in circuits relevant to aggression, potentially underlying the lack of aggression recovery.

(3) This is a continuation of my comment/concern directly above. If the induction of ara and arb aren't enough, then how can, as the authors state, androgen signaling be the primary driver of these behaviors?

In response to this follow-up comment, we would like to clarify that, as described above, the lack of aggression recovery in E2-treated cyp19a1b-deficient males is not due to insufficient induction of ara and arb, but instead is likely because esr2b was also induced in aggression-relevant circuits, which may have suppressed aggression. Therefore, the concern that androgen signaling cannot be the primary driver of these behaviors is not applicable.

(4) The authors' point about sticking with the terminology for the ar genes as "ara" and "arb" is not convincing. The whole point of needing a change to match the field of neuroendocrinology as a whole (that is, across all vertebrates) is researchers, especially those with high standing like the Okubo group, adopt the new terminology. Indeed, the Okubo group is THE leader in medaka neuroendocrinology. It would go a long way if they began adopting the new terminology of "ar1" and "ar2". I understand this may be laborious to a degree, and each group can choose to use their terminology, but I'd be remiss if I didn't express my opinion that changing the terminology could help our field as a whole.

We sincerely appreciate the reviewer’s thoughtful comments regarding nomenclature consistency in vertebrate neuroendocrinology. We understand the motivation behind the suggestion to adopt ar1 and ar2. However, we consider the established nomenclature of ara and arb to be more appropriate for the following reasons.

First, adopting the ar1/ar2 nomenclature would introduce a discrepancy between gene and protein symbols. According to the NCBI International Protein Nomenclature Guidelines (Section 2B.Abbreviations and symbols; https://www.ncbi.nlm.nih.gov/genbank/internatprot_nomenguide/), the ZFIN Zebrafish Nomenclature Conventions (Section 2. PROTEINS:https://zfin.atlassian.net/wiki/spaces/general/pages/1818394635/ZFIN+Zebrafish+Nomenclature+Conventions), and the author guidelines of many journal (e.g.,https://academic.oup.com/molehr/pages/Gene_And_Protein_Nomenclature), gene and protein symbols should be identical (with proteins designated in non-italic font and with the first letter capitalized). Maintaining consistency between gene and protein symbols helps avoid unnecessary confusion. The ara/arb nomenclature allows this, whereas ar1/ar2 does not.

Second, the two androgen receptor genes in teleosts are paralogs derived from the third round of wholegenome duplication that occurred early in teleost evolution. For such duplicated genes, the ZFIN Zebrafish Nomenclature Conventions (Section 1.2. Duplicated genes) recommend appending the suffixes “a” and “b” to the approved symbol of the human or mouse ortholog. This convention clearly indicates that these genes are whole-genome duplication paralogs and provides an intuitive way to represent orthologous and paralogous relationships between teleost genes and those of other vertebrates. As a result, it has been widely adopted, and we consider it logical and beneficial to apply the same principle to androgen receptors.

In light of these considerations, we respectfully maintain that the ara/arb nomenclature is more suitable for the present manuscript than the alternative ar1/ar2 system.

(5) In the discussion please discuss these potentially unexpected findings.(a) gal was unaffected in female cyp19a1 mutants, but they exhibit mating behaviors towards females. Given gal is higher in males and these females act like females, what does this mean about the function of gal/its utility in being a male-specific marker (is it one??)?(b) esr2b expression is higher in female cyp19a1 mutants. this is unexpected as well given esr2b is required for female-typical mating and is higher in females compared to males and E2 increases esr2b expression. please explain...well, what this means for our idea of what esr2b expression tell us.

We thank the reviewer for the insightful comments. As the female data have been removed from the manuscript, discussion of these findings in female cyp19a1b mutants is no longer necessary.

**Reviewer #3 (Recommendations For The Authors):**
The authors have addressed a number of answers to the reviewer's comments, notably they provided missing methodological information and rephrased the text. However, the authors have not addressed the main issues raised by the reviewers. Notably, it is regrettable that the reduced amount of brain aromatase cannot be confirmed, this seems to be the primary step when validating a new mutant. Even if protein products of the two genes may not be discriminated (which I can understand), it should be possible to evaluate the expression of a common messenger and/or peptide and confirm that aromatase expression is reduced in the brain. Since Cyp19a1b is relatively more abundant in the brain Cyp19a1a, this would strengthen the conclusion and provide confidence that the mutant indeed does silence aromatase expression in the brain. Although these short comings are acknowledged in the rebuttal letter, this is not mentioned in the discussion. Doing so would make the manuscript more transparent and clearer.

As noted in Response to reviewer #3’s comment 1 on weaknesses, we have validated the loss of Cyp19a1b function by measuring its transcript levels in the brain and predicting the three-dimensional structure of the mutant protein. These analyses confirmed that Cyp19a1b function is indeed lost, thereby increasing the reliability of our conclusions.

FigS1 - panels C&D please indicate in which tissue were hormones measured. Blood?

We thank the reviewer for pointing this out. In our study, “peripheral” refers to the caudal half of the body excluding the head and visceral organs, not blood. Accordingly, we have revised the figure legend and the description in the Materials and Methods section as follows:

Legend for Figure 1B (line 787) now reads: “Levels of E2, testosterone, and 11KT in the brain (A) and peripheral tissues (caudal half of the body) (B) of adult cyp19a1b^+/+^, cyp19a1b^+/−^, and cyp19a1b^−/−^ males (n = 3 per genotype).”

Materials and methods (line 431): The sentence “Total lipids were extracted from the brain and peripheral tissues (from the caudal half) of” has been revised to “Total lipids were extracted from the brain and from peripheral tissues, specifically the caudal half of the body excluding the head and visceral organs, of.”

Additional Alterations:

We have reformatted the text and supporting materials to comply with the journal’s Author Guidelines. The following changes have been made:

(1) Figures and supplementary files are now provided separately from the main text.

(2) The title page has been reformatted without any changes to its content.

(3) In-text citations have been changed from numerical references to the author–year format.

(4) Figure labels have been revised from “Fig. 1,” “Fig. S1,” etc., to “Figure 1,” “Figure 1—figure supplement 1,” etc.

(5) Table labels have been revised from “Table S1,” etc., to “Supplementary file 1,” etc.

(6) Line 324: The typo “is” has been corrected to “are”.

(7) Line 382: The section heading “Materials and Methods” has been changed to “Materials and methods” (lowercase “m”).

(8) Line 383: The Key Resources Table has been placed at the beginning of the Materials and methods section.

(9) Line 389: The sentence “Sexually mature adults (2–6 months) were used for experiments, and tissues were consistently sampled 1–5 hours after lights on.” has been revised to “Sexually mature adults (2–6 months) were used for experiments and assigned randomly to experimental groups. Tissues were consistently sampled 1–5 hours after lights on.”

(10) Line 393: The sentence “All fish were handled in accordance with the guidelines of the Institutional Animal Care and Use Committee of the University of Tokyo.” has been removed.

(11) Line 589: The following sentence has been added: “No power analysis was conducted due to the lack of relevant data; sample size was estimated based on previous studies reporting inter-individual variation in behavior and neural gene expression in medaka.”

(12) Line 598: The reference list has been reordered from numerical sequence to alphabetical order by author.

(13) In the figure legends, notations such as “A and B” have been revised to “A, B.”